# Arousal dependent modulation of thalamo-cortical functional interaction

Iain Stitt[1], Zhe Charles Zhou[1,2], Susanne Radtke-Schuller[1] & Flavio Fröhlich [1,2,3,4,5,6]

Ongoing changes in arousal influence sensory processing and behavioral performance. Yet the circuit-level correlates for this influence remain poorly understood. Here, we investigate how functional interaction between posterior parietal cortex (PPC) and lateral posterior (LP)/Pulvinar is influenced by ongoing fluctuations in pupil-linked arousal, which is a non-invasive measure of neuromodulatory tone in the brain. We find that fluctuations in pupil-linked arousal correlate with changes to PPC to LP/Pulvinar oscillatory interaction, with cortical alpha oscillations driving activity during low arousal states, and LP/Pulvinar driving PPC in the theta frequency band during higher arousal states. Active visual exploration by saccadic eye movements elicits similar transitions in thalamo-cortical interaction. Furthermore, the presentation of naturalistic video stimuli induces thalamo-cortical network states closely resembling epochs of high arousal in the absence of visual input. Thus, neuromodulators may play a role in dynamically sculpting the patterns of thalamo-cortical functional interaction that underlie visual processing.

[1] Department of Psychiatry, University of North Carolina at Chapel Hill, Chapel Hill, NC 27599, USA. [2] Neurobiology Curriculum, University of North Carolina at Chapel Hill, Chapel Hill, NC 27599, USA. [3] Department of Neurology, University of North Carolina at Chapel Hill, Chapel Hill, NC 27599, USA. [4] Department of Cell Biology and Physiology, University of North Carolina at Chapel Hill, Chapel Hill, NC 27599, USA. [5] Department of Biomedical Engineering, University of North Carolina at Chapel Hill, Chapel Hill, NC 27599, USA. [6] Neuroscience Center, University of North Carolina at Chapel Hill, Chapel Hill, NC 27599, USA. Correspondence and requests for materials should be addressed to F.F. (email: flavio_frohlich@med.unc.edu)

As exemplified by the daydreaming student drifting in and out of focus during class, the brain exhibits the ability to rapidly transition between states of varying engagement with the external world. Rather than resulting from changes in anatomical connections between neurons, such moment-to-moment variability in internal brain state arises through changes in network-level activity patterns that are constrained by the brains structural framework[1]. In this scheme, cognition and behavior emerge from the dynamic interaction of widely distributed, but functionally specialized cortical and subcortical brain regions[2,3].

In the visual system, higher order brain structures such as the lateral posterior (LP)/Pulvinar nuclear complex of the thalamus play an essential role in orchestrating the patterns of large-scale cortical interaction that underlie visual behavior[4–6]. Providing the structural framework for this coordinating role, LP/Pulvinar exhibits reciprocal anatomical connectivity with widely distributed visual cortical areas[6]. In particular, posterior parietal cortex (PPC), a higher-order association area that itself is a hub of sensory integration, sends dense projections to LP/Pulvinar[7]. Converging evidence suggests that the selective synchronization of neuronal oscillations between LP/Pulvinar and cortex

facilitates communication between inter-connected cortical sites, and that such patterns of thalamo-cortical dynamics reflect the circuit-level computations that underlie visual sensory processing and behavior[2,8]. Yet, despite the established importance of thalamo-cortical interaction for visual processing and behavior[4,5], we still lack a clear understanding of how the relative strength of thalamus-to-cortex and cortex-to-thalamus signals are dynamically tuned to meet current behavioral demands.

Recent theoretical work proposed neuromodulators as a mechanism for modifying information flow in neuronal networks on a short temporal scale[9]. Indeed, the neuromodulators noradrenaline and acetylcholine modulate the intrinsic properties of neurons across both cortical and thalamic areas[10–15]. Beyond the effects of such neurotransmitters on the local cellular level in cortex and thalamus, it remains unclear if neuromodulators help to shape emergent patterns of information routing in thalamo-cortical networks. Recent findings have linked ongoing fluctuations in pupil diameter to the release of noradrenaline and acetylcholine from synaptic terminals in the cortex[16], indicating that non-invasive measurement of pupil-linked arousal enables the indirect inference of neuromodulatory tone in the brain.

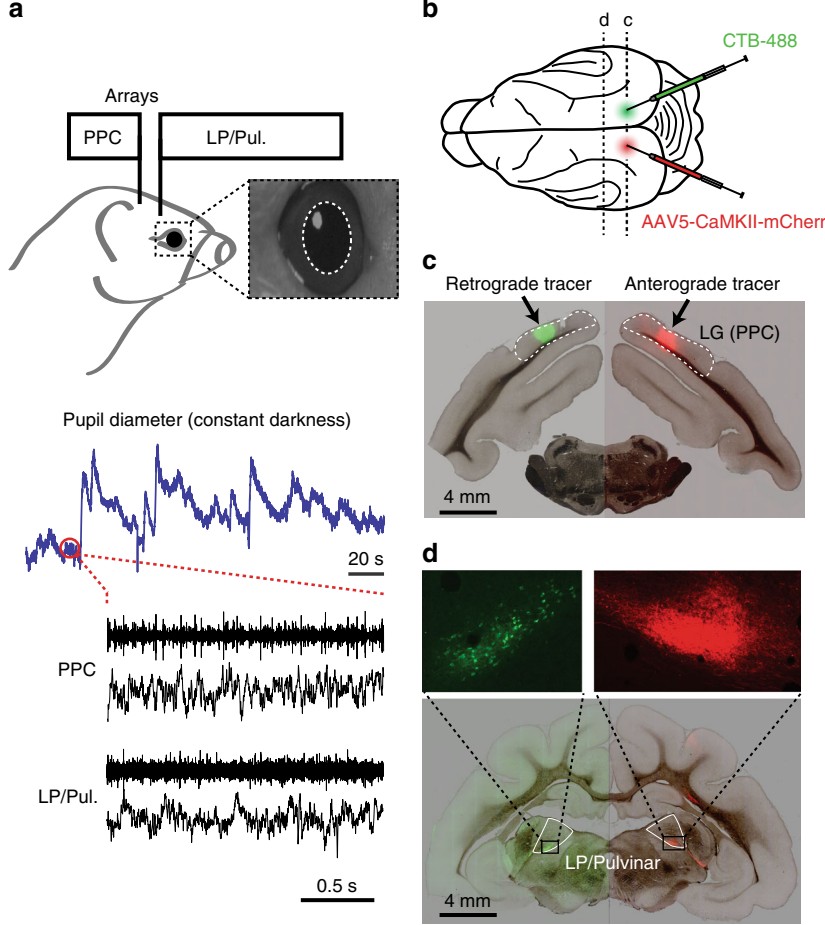

**Fig. 1** Experimental setup and anatomical connectivity between LP/Pulvinar and PPC. **a** Diagram illustrating how neural signals from PPC and LP/Pulvinar were simultaneously recorded with pupil diameter. Inset image shows a typical view of the ferret infrared eye tracking, with the pupil outlined in white. Below are raw traces of ongoing fluctuations pupil diameter and co-recorded spiking and LFP activity in PPC and LP/Pulvinar. Note that pupil diameter spontaneously fluctuates on both short and long timescales. **b** Anterograde (rAAV5-CaMKII-mCherry) and retrograde (CTB-488) tracers were injected into PPC in the left and right hemispheres, respectively. **c** Brightfield image of a brain section containing the PPC injection sites overlaid with green and red fluorescence channels. Fluorescent blobs show the location of anterograde and retrograde tracer in PPC. **d** Brightfield image of thalamus overlaid with fluorescence from red and green channels. Retrograde labeling of cell bodies (green) and anterograde labeling of axonal projections (red) in corresponding locations of LP/Pulvinar illustrate reciprocal connectivity between PPC and LP/Pulvinar in the ferret. LG lateral gyrus, PPC posterior parietal cortex

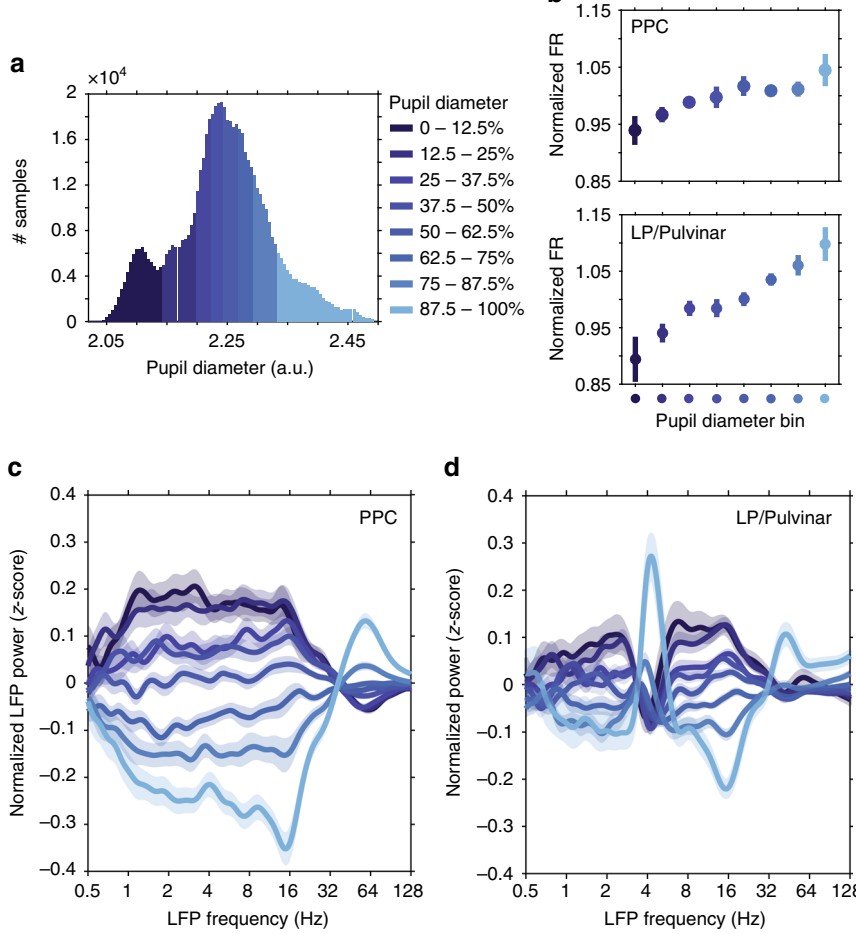

**Fig. 2** Neuronal spiking rate and LFP spectral power in PPC and LP/Pulvinar are modulated with pupil diameter. **a** A representative example of how pupil diameter time series were divided up into bins (each bin represents 12.5% of all samples). Neurophysiological data were then analyzed according to pupil diameter bin. **b** The normalized spiking rate in both PPC and LP/Pulvinar increases with pupil dilation. Color denotes pupil diameter, as indicated in **a**. **c** The mean (±SEM) z-scored LFP power in PPC as a function of carrier frequency across pupil diameter bins. Low (<30 Hz) and high (>30 Hz) frequency LFP power displayed opposing relationships to pupil diameter. **d** the same as **c** but for LP/Pulvinar LFP power. Note the antagonism between low- and high-frequency LFP oscillatory power is similar to PPC, with the exception of the theta band (~4 Hz), which displays increased power during large pupil diameter states

To investigate the role of neuromodulation in shaping thalamo-cortical functional interaction, we monitored ongoing fluctuations in pupil-linked arousal in awake head-restrained ferrets, while simultaneously recording spiking and local field potential (LFP) activity from LP/Pulvinar and PPC (Fig. 1a). We found that the carrier frequency of thalamo-cortical synchronization varied with arousal, and that the direction of causal interaction between cortex and thalamus switched between low and high arousal levels. Moreover, we show that such transitions in network dynamics are not exclusive to ongoing activity in relative absence of visual input, but also occur when animals are actively engaged in sensory processing. We suggest that neuromodulators shape thalamo-cortical functional interaction by altering the relative contribution of thalamic and cortical signals to emergent network dynamics.

## Results

### Reciprocal connectivity between PPC and LP/Pulvinar.
Functional interaction between brain regions is constrained by structural connectivity. To map the precise anatomical connectivity of the regions of interest in this study, we injected anterograde (AAV5-CaMKII-mCherry) and retrograde (CTB-488) tracers into the left and right PPC, respectively (Fig. 1b, c). PPC injections were made at locations that corresponded to the site of cortical multielectrode array implantation in other animals (Supplementary Fig. 1). We observed anterograde labeled fibers in the ventral portion of LP/Pulvinar (Fig. 1d), indicating that PPC neurons send projections to this sub region of the thalamus. In addition, we observed retrograde labeled cell bodies at the corresponding location in the opposite hemisphere (Fig. 1d), indicating that LP/Pulvinar also projects back to the location of the injection site in PPC. These results indicate that regions of thalamus and cortex where electrophysiological recordings were obtained display reciprocal connectivity, establishing the physical substrate for studying how thalamo-cortical interaction varies with pupil-linked arousal.

### Arousal-dependent changes in neuronal spiking and LFP.
Given that both PPC and LP/Pulvinar receive dense projections from brainstem neuromodulatory systems[17], we first examined how firing rate was modulated with pupil-linked arousal. Large and small pupil sizes indicated high and low arousal states, respectively. Consistent with previous in vitro and in vivo work on the influence of noradrenaline on neuronal excitability[10–12,14,15,18–20], we found that neuronal firing rate at the population level in both PPC and LP/Pulvinar significantly increased with pupil diameter (Fig. 2b

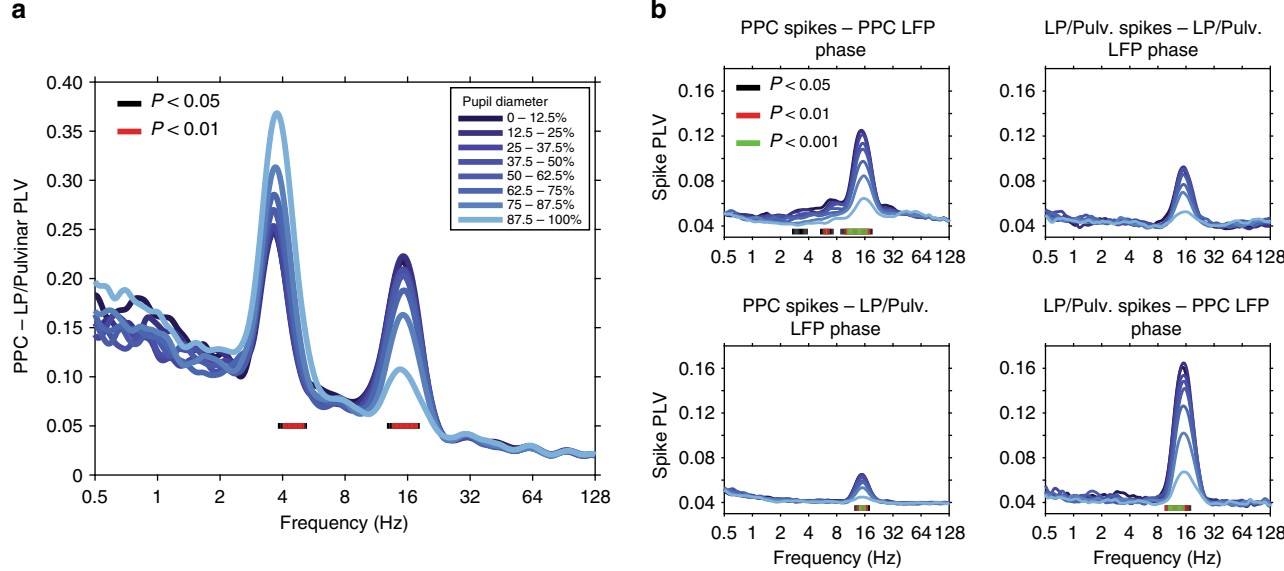

**Fig. 3** Thalamo-cortical synchronization varies with ongoing fluctuations in pupil-linked arousal. **a** PLV measured between LP/Pulvinar and PPC as a function of LFP frequency and pupil diameter. Pupil diameter is denoted by color (see legend). Note the prominent thalamo-cortical phase synchronization in the theta (~4 Hz) and alpha (12–17 Hz) carrier frequency bands. Significant modulation across pupil diameter bins is indicated by bars plotted below PLV traces (one-way ANOVA, FDR-corrected $P$-values). PLV in the theta band significantly increased with pupil dilation, while alpha PLV significantly decreased with pupil dilation. **b** The phase synchronization (PLV) of spiking activity to LFP rhythms recorded within the same brain structure (top row), as well as between regions (bottom row). Spike PLV both locally within PPC and LP/Pulvinar, as well as between PPC and LP/Pulvinar revealed phase locking of spiking activity to alpha oscillations. Significant modulation of spike PLV across pupil diameter bins is indicated by color bars plotted below PLV traces (one-way ANOVA, FDR-corrected $P$-values). Note that the strongest alpha band spike PLV was observed between LP/pulvinar spikes and PPC LFP phase

population, PPC: $P = 0.003$, one-way ANOVA, $r = 0.30$; LP/Pulvinar: $P = 9.56 \times 10e^{-11}$, $r = 0.55$). Multi-unit spiking activity in PPC could be more generally separated into three groups based on correlation with pupil diameter (Supplementary Fig. 2); units that increased firing rate with pupil dilation (40.3% of units, $n = 209/519$), units that decreased firing rate with pupil dilation (26.0% of units, $n = 135/519$), and units that showed no significant correlation with pupil diameter (33.7% of units, $n = 175/519$). Such diversity in PPC spiking activity related to arousal is in general agreement with work in mouse visual and frontal cortices, which found that subpopulations of neurons in mouse visual cortex either increased or decreased firing rate with arousal[21,22]. In contrast to PPC, the majority of multi-units in LP/Pulvinar displayed increased firing rate with pupil diameter (72.2%, $n = 122/169$), with only a negligible portion of units displaying a decrease in firing rate (3.0%, $n = 5/169$). Together, these results suggest greater heterogeneity of spiking dependence on arousal in cortex, and may reflect the greater complexity and diversity of neuronal populations that comprise cortical circuits[22].

To examine how signatures of local network dynamics were altered by arousal, we computed changes in LFP power spectra as a function of pupil diameter (Fig. 2c, d). Pupil diameter-related changes in LFP power were characterized by opposing effects on low- (<30 Hz) and high-frequency (>30 Hz) LFP oscillations in PPC (for raw power spectra, see Supplementary Fig. 3); low frequency power was stronger during small pupil diameter epochs, while high frequency power was stronger during large pupil diameter epochs (Fig. 2c). LP/Pulvinar displayed similar pupil size-dependent antagonism between low- and high-frequency LFP signals, with the exception of LFP power in the theta band (3.3–4.5 Hz), which displayed peak power during large pupil diameter epochs (Fig. 2d). In both PPC and LP/Pulvinar, large pupil diameter epochs were associated with reduced LFP power in the 12–17 Hz frequency range. Since LFP signal power and thalamo-cortical coherence in 12–17 Hz frequency range was reduced during visual stimulation (Supplementary Fig. 3), we

posit that rhythms in this frequency band represents a homolog of the alpha oscillation in the ferret.

**Modulation of thalamo-cortical synchronization with arousal.** Considering that complex behaviors such as attention require the coordination of neural activity between cortex and thalamus[5], we next asked if signatures of thalamo-cortical functional interaction varied with ongoing fluctuations in pupil-linked arousal. We observed thalamo-cortical LFP phase synchronization, measured by phase locking value (PLV) in the theta and alpha carrier frequency bands (Fig. 3a, phase delay in theta band = 9.6 ms, 0.55 circular variance; phase delay in alpha band = 30.2 ms, 0.78 circular variance). Strikingly, the strength of phase synchronization in these frequency bands was modulated in opposing directions by changes in pupil diameter, with theta phase synchronization significantly increasing ($P < 0.01$, one-way ANOVA), and alpha phase synchronization significantly decreasing with pupil dilation (Fig. 3a, $P < 0.01$, one-way ANOVA; see Supplementary Fig. 4 for power-matched PLV analysis).

These results suggest that fluctuations in pupil-linked arousal coincide with a shift in the carrier frequency of thalamo-cortical functional interaction from the alpha band in low arousal states, to the theta band in high arousal states. In line with this hypothesis, spike cross-correlation analysis uncovered synchronous oscillatory patterns of thalamo-cortical neuronal firing occurring in the alpha band (Supplementary Fig. 5). Transition from small to large pupil diameter was marked by a significant decrease in oscillatory spike correlations in the alpha band (Supplementary Fig. 5c, $P = 4.1e^{-6}$, one-way ANOVA, $r = -0.42$). In addition to LFP–LFP and spike–spike correlations, we also observed spike–LFP phase synchronization in the alpha frequency band both locally within LP/Pulvinar and PPC, as well as between thalamus and cortex (Fig. 3b). Spike–LFP phase synchronization in the alpha band significantly decreased with pupil dilation locally within PPC as well as between regions

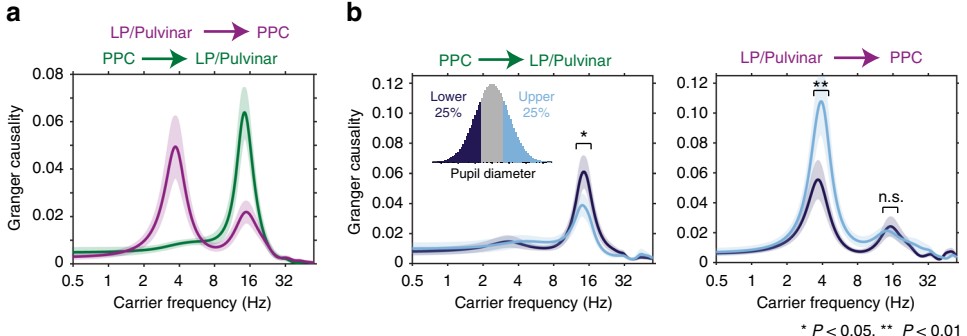

**Fig. 4** Arousal level determines the direction and carrier frequency of thalamo-cortical causal interaction. **a** Spectrally resolved Granger causality shows the carrier frequencies of directed interaction from LP/Pulvinar to PPC (magenta), and PPC to LP/Pulvinar (green, ±SEM). LP/Pulvinar has a causal influence on PPC in the theta and alpha frequency bands. In the opposing direction, PPC has a causal influence on LP/Pulvinar in the alpha band. **b** Granger causality was measured for time periods where the pupil diameter was small (<25%, dark blue) and large (>25%, light blue), respectively. Data were subsampled to match power distributions between conditions. The causal influence of PPC alpha oscillations on LP/Pulvinar was significantly stronger during small pupil diameter epochs (left plot, $P = 0.018$, $t$-test). In contrast, the causal influence of LP/Pulvinar theta oscillations on PPC was significantly greater during large pupil diameter epochs (right plot, $P = 0.0015$, $t$-test)

($P < 0.001$, one-way ANOVA). Consistent with previous work in humans[23], peak alpha frequency significantly increased in PPC with arousal (Supplementary Fig. 6, $P = 0.003$, one-way ANOVA, $r = 0.32$); however, LP/Pulvinar displayed no such relationship. In contrast to results in the alpha band, spike–spike correlation and spike–LFP phase synchronization were weak in the theta frequency band both locally and between regions (Fig. 3b). Together, these results suggest that alpha rhythms synchronize thalamo-cortical network activity in low arousal states, with increasing arousal leading to the desynchronization of spiking activity and a transition to the theta carrier frequency for thalamo-cortical functional interaction.

**Arousal-dependent thalamo-cortical effective connectivity.** Given that we observed two distinct carrier frequencies of thalamo-cortical functional interaction that were differentially modulated by arousal, we asked if activity in these frequency bands reflected directed interaction between LP/Pulvinar and PPC. We quantified directionality in thalamo-cortical interaction by computing spectrally resolved Granger causality and performing phase slope index (PSI) analyses between LP/Pulvinar and PPC LFP signals (Fig. 4a; see Supplementary Fig. 7 for PSI analysis). In line with previous results[5], we found significant reciprocal causal interaction between PPC and LP/Pulvinar in the alpha frequency band ($P < 0.001$, permutation test). However, in contrast to a previous report[5], we found that the cortical influence on thalamus was significantly stronger than thalamic influence on cortex in the alpha band ($P = 0.003$, $t$-test). PSI analyses confirmed this result (Supplementary Fig. 7), with a larger proportion of cortical channels driving thalamus (32%) than vice versa (11%). Both Granger causality and PSI findings are consistent with the hypothesis that alpha rhythms arise from reciprocal interaction between thalamic and cortical alpha oscillators. In contrast to alpha rhythms, Granger causal interaction in the theta band was only observed from LP/Pulvinar to PPC ($P < 0.001$, permutation test; Fig. 4a). PSI analyses agreed with this result, with 19% of channel pairs displaying significant thalamus-to-cortex interaction, while only 3% displayed significant cortex-to-thalamus interaction (Supplementary Fig. 7). Collectively, these results suggest that theta rhythms predominantly propagate from thalamus to cortex, while alpha rhythms propagate principally from cortex to thalamus.

To examine how thalamo-cortical effective connectivity is modulated by arousal, we recomputed Granger causality for epochs that represented the lower and upper 25% of pupil

diameter for each recording, respectively. Since LFP power was highly dependent on arousal, we subsampled data to match power distributions across pupil diameter bins (Supplementary Fig. 8). The causal influence of PPC on LP/Pulvinar in the alpha band was significantly stronger for small pupil diameter states ($P = 0.018$, $t$-test, Fig. 4b, left). In contrast, LP/Pulvinar causal influence on PPC in the theta band was significantly stronger in large pupil diameter states ($P = 0.0015$, $t$-test, Fig. 4b, right). These results indicate that fluctuations in pupil-linked arousal result in a dynamic switch in the predominant carrier frequency of thalamo-cortical communication and the direction of causal interaction between LP/Pulvinar and PPC.

**Thalamo-cortical network dynamics during visual processing.** Until now, we have examined how variability in thalamo-cortical network dynamics relate to ongoing fluctuations in pupil-linked arousal in the absence of visual input. Are such patterns of thalamo-cortical network dynamics exclusive to recordings in the dark, or are they reflective of more generalizable network states that also emerge during periods when animals are engaged in sensory processing? To answer this question, we investigated how patterns of activity in LP/Pulvinar and PPC are modulated in response to the presentation of naturalistic images or videos (Fig. 5a). Naturalistic visual stimuli elicited increased gamma power and decreased alpha power in both LP/Pulvinar and PPC (Fig. 5b, c for videos; see Supplementary Fig. 9 for images). In general, naturalistic video stimuli elicited more robust spiking and LFP responses than static images (PPC static image firing rate = $13.45 \pm 1.65$, video firing rate = $14.33 \pm 1.64$, $P = 8.8e^{-5}$, sign test; LP/Pulvinar static image firing rate = $8.72 \pm 1.04$, video firing rate = $9.68 \pm 1.09$, $P = 8.0e^{-7}$, sign test). In the prestimulus period we found thalamo-cortical LFP synchronization at theta and alpha carrier frequencies (Fig. 5d), similar to small pupil diameter states in the dark. However, upon presentation of video stimuli, thalamo-cortical synchronization rapidly shifted to the theta carrier frequency, with an accompanying reduction in alpha synchronization. LFP power modulations during visual processing were significantly correlated with LFP power spectra during large pupil diameter states in the absence of any stimulus (PPC correlation per animal: 0.51*, 0.94*, 0.92*, 0.82*; LP/Pulvinar correlation per animal: 0.13, 0.66*, 0.06, 0.54*; *$P < 0.001$). Theta synchronization maintained throughout the duration of stimulus presentation, before returning to the theta and alpha carrier frequencies after stimulus offset (Fig. 5d). Granger causality analysis

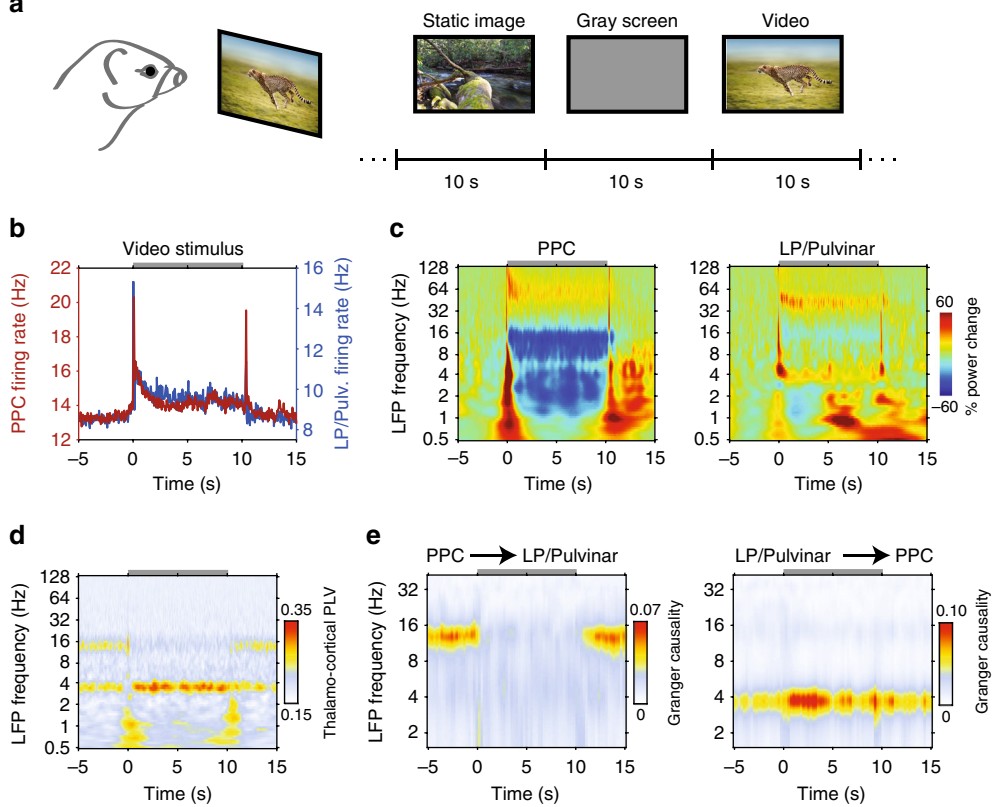

**Fig. 5** Visual processing induced changes in thalamo-cortical network dynamics. **a** Animals passively viewed a collection of naturalistic images or videos. During the interstimulus interval a gray screen was presented. The image "Running Cheetah" by Freder is licensed under the Standard iStock Photo License (Getty Images). **b** Population mean spike rate in PPC and LP/Pulvinar during presentation of video stimuli. The gray bar at the top of the plot indicates the duration of stimulation. **c** Population LFP spectrograms from PPC (left) and LP/Pulvinar (right) during presentation of naturalistic videos. LFP power was normalized to the period −5 to −1 s before stimulus onset. Note the decrease in alpha oscillatory power in PPC during stimulus presentation. **d** Across-session average thalamo-cortical phase synchronization in response to naturalistic video stimuli. PLV in the theta band is elevated during stimulus presentation, while alpha PLV is weaker. **e** Time and frequency resolved Granger causality analysis computed between PPC and LP/Pulvinar LFP signals for naturalistic video stimuli. The onset of video stimuli leads to a breakdown of PPC causal influence on LP/Pulvinar in the alpha band (left plot), and an increase of LP/Pulvinar causal influence on PPC in the theta frequency band (right plot)

confirmed that video stimuli induce a rapid reversal in the direction and carrier frequency of thalamo-cortical effective connectivity similar to what we observe during small and large pupil diameter epochs in the dark (Fig. 5e). These findings suggest that during presentation of video stimuli the LP/Pulvinar–PPC network shifts towards a state defined by thalamus-driven theta oscillations. In addition, these results illustrate that arousal-dependent thalamo-cortical network states that spontaneously occur in the dark are also recruited when animals are processing sensory information.

**Pupil diameter correlates with saccade and network dynamics.** What is the link between the emergence of these patterns of thalamo-cortical network dynamics and behavior? Given the well-established link between saccadic eye movements and cognitive processes[24], we answer this question by examining how thalamo-cortical network dynamics relate to saccade behavior. Saccade kinetics in the ferret displayed similar characteristics to saccades in humans[25] and non-human primates[26] (Fig. 6a, Supplementary Fig 10), albeit with a lower overall saccade rate ($0.32 \pm 0.03$ Hz). Animals displayed an elevated rate of saccades while viewing video stimuli ($0.35 \pm 0.03$ saccades/s videos stimuli, $0.19 \pm 0.02$ saccades/s prestimulus, $P = 2.7e^{-7}$ t-test, Fig. 6b), illustrating that ferrets utilize saccades to actively sample the visual environment.

Animals also performed saccades in the dark (Supplementary Fig. 10), where saccade rate was significantly modulated by ongoing fluctuations in pupil diameter (Fig. 6c, $P = 3.80e^{-15}$, one-way ANOVA). Indeed, saccade-triggered analysis of pupil diameter in the dark revealed significant pupil dilations prior to the onset of saccadic eye movements ($P = 0.004$, t-test, Fig. 6d). This result suggests that oculomotor behavior may arise from a more general shift towards an aroused state. In line with this, PPC and LP/Pulvinar LFP power during saccades was significantly correlated with power spectra during large pupil diameter states (PPC correlation per animal: 0.95*, 0.99*, 0.91*, 0.94*; LP/Pulvinar correlation per animal: 0.71*, 0.91*, 0.81*, 0.88*; *$P < 0.01$), with power modulations occurring over a time course of several seconds around saccades (Fig. 6e, $P < 0.05$, test against random saccade times, Supplementary Figure 11a, b). Similarly, thalamo-cortical synchronization in the alpha band tended to decrease during saccades, while theta band synchronization tended to increase (Fig. 6f); however, these changes were significant in a minority of recordings ($P < 0.05$, Supplementary Figure 11c). To disentangle the role of saccades and pupil-linked arousal in modifying thalamo-cortical network dynamics, we repeated functional connectivity analyses after removing peri-saccadic epochs. Overall, we found that pupil-linked arousal based analyses reflected results prior to removal of peri-saccadic epochs (Supplementary Figure 12), with the exception of alpha LFP synchronization, which did not reach significance.

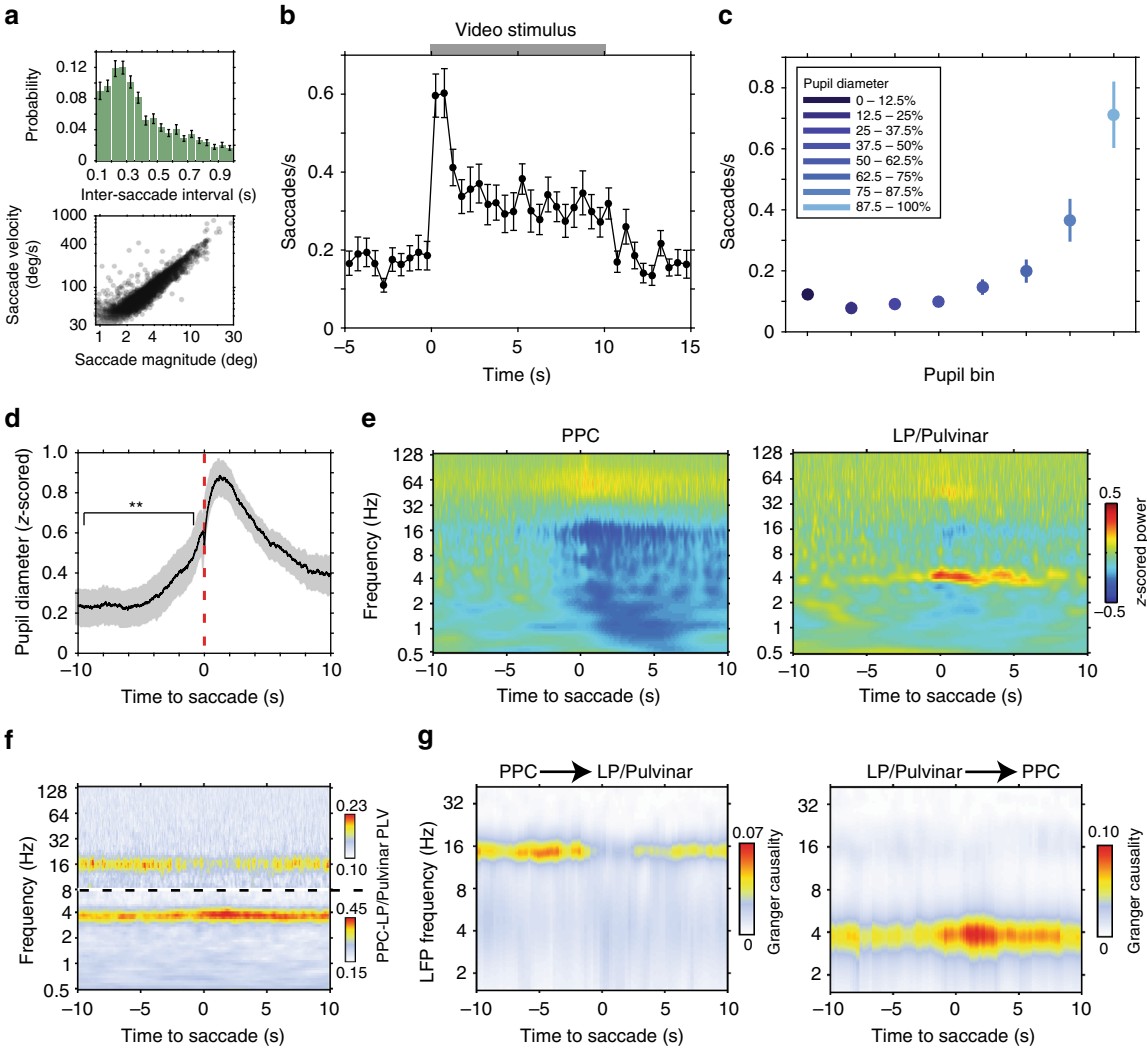

**Fig. 6** Saccades link visual sensory processing and pupil-linked arousal-related changes in thalamo-cortical dynamics. **a** Distribution of inter saccade interval (top, ±SEM) shows that ferrets actively sample the visual environment rhythmically. The relationship between saccade magnitude and peak velocity (bottom) reflects the ballistic nature of saccades in ferrets. **b** Saccade rate during naturalistic video presentation illustrates that animals actively sample visual stimuli by showing an elevated rate of saccades ($n = 26$ sessions). **c** Saccade rate as a function of pupil diameter in the dark. The rate of saccades during large pupil diameter states is comparable to the rate of saccades when animals are actively sampling naturalistic videos. **d** Mean (±SEM) fluctuations in pupil diameter time locked to saccadic eye movements in the dark. Transient increases in pupil diameter precede saccades ($P = 0.004$, $t$-test). **e** Population average LFP power spectrograms in PPC (left) and LP/Pulvinar (right) time locked to saccades in the dark. LFP power was $z$-score normalized across the entire recording session. **f** Population average PPC to LP/Pulvinar PLV time locked to saccades in the dark. Dotted line indicates a break in the color scale, as shown to the right of the figure. **g** Across-session average time and frequency resolved Granger causality between PPC and LP/Pulvinar around the occurrence of saccades in the dark. Active sampling by saccades was associated with a decrease in PPC causal influence on LP/Pulvinar in the alpha band and an increase in LP/Pulvinar on PPC in the theta band

To test if the direction of thalamo-cortical interaction also changed during saccade behavior, we computed spectrally resolved Granger causality analysis time-locked to saccades (Fig. 6g). Saccades were associated with a significant decrease in PPC causal influence on LP/Pulvinar in the alpha band, accompanied by a significant increase of LP/Pulvinar causal influence on PPC in the theta band ($P < 0.05$, test against random saccade times, Supplementary Figure 11d). These results strongly support the hypothesis that dynamic changes in thalamo-cortical causal interaction facilitate active visual sampling of the environment.

How do ongoing fluctuations in thalamo-cortical network dynamics affect the way animals sample visual stimuli? To answer this question, we correlated the number saccades performed during each trial of visual stimulus presentation with the strength

of thalamo-cortical synchronization in the theta and alpha frequencies bands (Fig. 7a). We found a weak but significant positive correlation between synchronization in the theta band and the number of saccades performed per trial for both video and image stimulus conditions (Fig. 7a for video stimuli, $P = 0.0002$, $R = 0.17$; for image stimuli see Supplementary Fig. 13, $P = 0.0009$, $R = 0.15$). In contrast, we found a significant negative correlation between synchronization in the alpha band and the number of saccades per trial (video stimuli $P = 0.011$, $R = -0.11$; image stimuli $P = 0.0001$, $R = -0.17$). Thus, the state of the thalamo-cortical system as defined by its oscillatory functional connectivity indexed the level of engagement with the external world.

Finally, to show that fluctuations in pupil-linked arousal also affect how animals process incoming sensory information, we

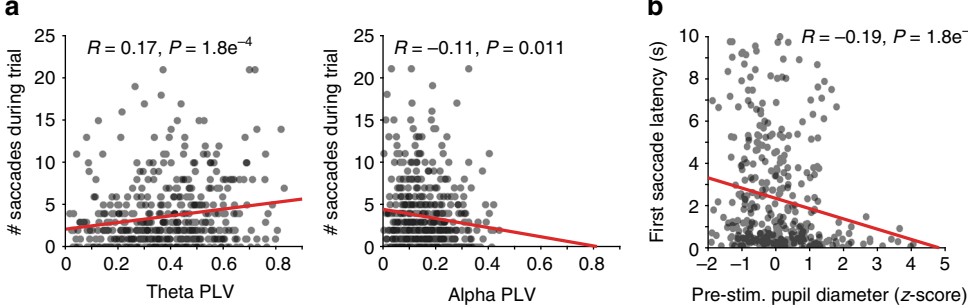

**Fig. 7** Thalamo-cortical synchronization and pupil-linked arousal correlate with saccade behavior. **a** Correlation of thalamo-cortical phase synchronization in the theta (left) and alpha (right) carrier frequency bands and the number of saccades performed during presentation of naturalistic video stimuli. Theta PLV displays a significant positive correlation with saccadic sampling of stimuli, whereas alpha PLV displays a significant negative correlation. **b** Correlation of prestimulus pupil diameter to the latency of the first saccade for subsequent naturalistic video stimulus presentation. Significant negative correlation illustrates that animals sample visual stimuli more rapidly when they are in a more aroused state

correlated the prestimulus pupil diameter with the latency of the first saccade for each trial. We found a small but significant negative correlation between prestimulus pupil diameter and first saccade latency for both image and video stimuli (Fig. 7b, $P = 0.0001$, $R = -0.19$ for video stimuli; for images see Supplementary Fig. 13b, $P = 6.39e^{-5}$, $R = -0.22$), indicating that animals sampled more rapidly when incoming sensory input arrived during a state of heightened arousal. Taken together, these results show that ongoing fluctuations in pupil-linked arousal and associated changes in thalamo-cortical functional interaction affect the way in which animals sample the external environment with saccadic eye movements.

**Fluctuations in pupil-linked arousal and network dynamics.** Recent work has shown that faster pupil dilations are correlated with noradrenergic input to cortex, whereas slower fluctuations are correlated with cholinergic input[16]. To disentangle how thalamo-cortical network dynamics change with fast and slow fluctuations in pupil diameter, we computed the derivative of pupil diameter time series and repeated functional connectivity analyses based on pupil derivative (Fig. 8a). In contrast to the monotonic profile of LFP power changes with respect to pupil diameter (Fig. 2c, d), pupil derivative based analyses of LFP power, LFP synchronization, and spike–LFP synchronization were characterized by distinctive 'U' and inverted 'U' shaped curves (Fig. 8b–d). Rapid positive and negative deflections in pupil diameter correlated with strong thalamo-cortical synchronization in the theta band and weak synchronization in the alpha band (Fig. 8c). Conversely, epochs of more constant pupil diameter were characterized by weak thalamo-cortical synchronization in the theta band, and strong LFP and spike–LFP synchronization in the alpha band (Fig. 8c, d). Analysis of spike rate, LFP power, and saccade rate time-locked to transient pupil dilations revealed the fast timescale on which thalamo-cortical network dynamics and saccade behavior reorganized following rapid fluctuations in pupil diameter (Supplementary Fig. 14). Together, these results illustrate that fast and slow fluctuations in pupil-linked arousal correlate with distinct signatures of thalamo-cortical functional interaction, and are consistent with a role for different neuromodulatory subsystems modifying thalamo-cortical network dynamics over fast and slow timescales.

## Discussion
Due to their reciprocal connectivity with widely distributed cortical areas, higher-order thalamic structures are proposed to play an important role in orchestrating patterns of large-scale cortico-

cortical and thalamo-cortical interaction that underlie cognition[4]. Our results provide the first evidence that pupil-linked arousal, and by extension neuromodulation, play an important role in dynamically sculpting these patterns of thalamo-cortical functional interaction. We demonstrate that ongoing fluctuations in pupil-linked arousal lead to dynamic switching of both the direction and carrier frequency of thalamo-cortical communication, with low arousal states marked by cortical alpha oscillations driving synchronized activity between LP/Pulvinar and PPC, while higher arousal states were marked by LP/Pulvinar driving PPC in the theta frequency band. Furthermore, we show similar transitions in thalamo-cortical network dynamics during visual processing, and in particular, during active sampling of the external environment via saccades.

What is the functional role of cortically driven alpha synchronization in thalamo-cortical networks? One prominent hypothesis is that alpha oscillations reflect the precise temporal parsing of cortical activity via phasic inhibition[27]. Under this framework, layer 5 projection neurons in PPC entrain local alpha oscillations in the LP/Pulvinar through pulsed volleys of action potentials. Given that LP/Pulvinar projections to superficial layers of early visual cortex influence cortical state[28,29], PPC to LP/Pulvinar synchronization in the alpha band may act to gate or suppress the processing of incoming sensory information at early stages of visual cortex. Beyond this gating role, alpha oscillations have also been associated with cognitive processes that require internalization of attention, such as working memory[27,30] and creativity[31]. In agreement with the internalization of attention, we consistently observed reduced saccade behavior during periods of elevated alpha oscillations. Similar to humans[32], this alpha-dominant mode of network dynamics was disrupted by the presentation of visual stimuli. Interestingly, we found that the degree to which animals actively sampled stimuli with saccades was negatively correlated with thalamo-cortical synchronization in the alpha band. Thus, the dynamic switching away from the synchronized alpha oscillation network state may be a signature of the transition between internalized and overt processes of attention.

We note that the endogenous alpha frequency identified here does not match the classic 10 Hz rhythm reported in humans. Indeed, we favor a definition of neural rhythms based on underlying physiological mechanisms, as opposed to the arbitrary assignment of bands based on carrier frequency. Given that alpha oscillations are proposed to arise through reciprocal thalamo-cortical interaction[33], it is unsurprising that these rhythms exhibit a shorter period in animals with smaller brains (and therefore shorter conduction delays), such as dogs[34], cats[35], and ferrets[36].

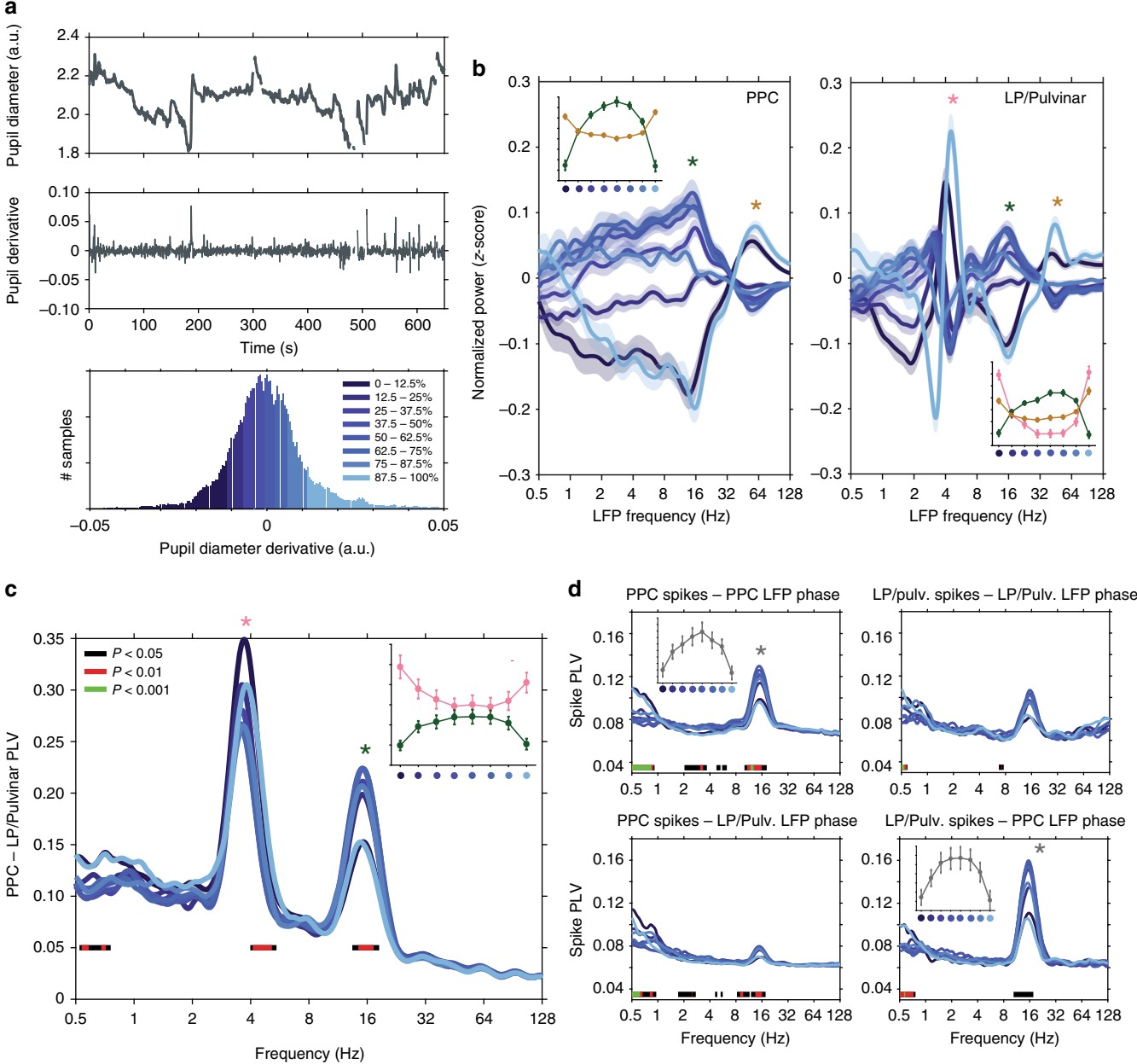

**Fig. 8** Rapid fluctuations in pupil diameter correlate with the reorganization of thalamo-cortical functional connectivity. **a** Raw pupil diameter (top) and pupil diameter derivative (middle) traces for one example recording (blinks have been removed from trace), and a histogram of pupil derivative time series broken into eight bins of equal size (bottom). Dark colors correspond to rapid pupil constriction, light colors correspond to rapid pupil dilation, and intermediate colors correspond to epochs of little change in pupil diameter. **b** The mean (±SEM) z-scored LFP power in PPC (left) and LP/Pulvinar (right) as a function of carrier frequency across pupil derivative bins. The insets in each figure illustrate how LFP power in theta, alpha, and gamma frequency bands changes across pupil derivative bins (frequency band indicated by asterisk). **c** PLV measured between LP/Pulvinar and PPC as a function of LFP frequency and the derivative of pupil diameter. **d** The phase synchronization (PLV) of spiking activity to LFP rhythms recorded within the same brain structure (top row), as well as between regions (bottom row) as a function of pupil derivative. Significant modulation of PLV and spike PLV across pupil derivative bins is indicated by color bars plotted below traces (one-way ANOVA, FDR-corrected P-values). Note the distinctive 'U' and inverted 'U' shape of many inset figures, indicating alpha synchronization/power is maximized during epochs of constant pupil diameter, and theta synchronization/power is maximized during epochs of rapid change in pupil diameter

In contrast to states of low arousal, thalamus-driven theta oscillations during states of high arousal were associated with an increase in saccade behavior. We found that transitions between alpha- and theta-dominant modes of thalamo-cortical synchronization arose transiently around the occurrence of saccades. These data suggest that the causal influence of thalamus on cortex in the theta band may play a role in active visual sampling and

overt attention. Lending support to this, previous studies in primates found that cortical LFP signals synchronize to microsaccades occurring rhythmically at the theta frequency[26]. Furthermore, theta oscillations temporally coordinate gamma band synchronization between lower and higher order visual cortices during attention allocation[37]. Therefore, the thalamic causal influence on cortex in the theta band may represent a

mechanism to selectively synchronize distributed visual cortical regions[5,6], facilitating visual exploration and attentional selection. Given that LP/Pulvinar is reciprocally connected with widely distributed visual cortical regions, it certainly exhibits the anatomical framework to play such a role in orchestrating cortical network dynamics and information routing[6]. However, we observed only weak locking of spiking activity to theta oscillation phase in LP/Pulvinar, suggesting the inputs that synchronize LP/Pulvinar and PPC theta oscillations are likely subthreshold and do not predominantly originate from the population of LP/Pulvinar neurons we recorded from. Alternatively, these theta-synchronizing inputs may originate from other brain regions such as the prefrontal cortex[38].

Fluctuations in pupil diameter under constant luminance have been used as a general measure of arousal and cognitive load[39]. A common assumption regarding the relationship between pupil diameter and arousal was that the locus coeruleus (LC), the main source of noradrenergic neuromodulatory input to the forebrain, must somehow form part of the brainstem circuit that controls pupil motility. Indeed, several invasive electrophysiological studies in monkeys have shown that fluctuations in pupil diameter under constant luminance are correlated to neural activity in the LC[40,41]. Further work in humans showed that pupil diameter fluctuations correlate with blood-oxygenation-level-dependent signals localized to the LC[42,43]. However, correlations were generally weak, and no direct anatomical connection linking the LC and brainstem pupil motility nuclei could be found to explain such correlations[44,45]. In a recent study Reimer et al.[16] showed that fluctuations in pupil diameter track both noradrenergic and cholinergic input to cortex, with more rapid pupil dilations reflecting changes in noradrenergic input, and longer-lasting fluctuations reflecting the sustained activity of cholinergic synapses. Considering we observe distinct patterns of thalamo-cortical network activity and saccade behavior related to fast and slow fluctuations in pupil diameter (Supplementary Fig. 12), we postulate that the combined actions of neuromodulatory subsystems play a crucial role in shaping thalamo-cortical network dynamics and oculomotor behavior.

How do ongoing fluctuations in neuromodulatory tone alter large-scale network interactions in the brain? Previous in vitro work has classified in detail the effects of various neuromodulatory subsystems on intrinsic cellular properties[10–12,14,18] and the generation of rhythmic activity in local circuits[46]. However, these local cellular changes do not explain the dynamic rerouting of thalamo-cortical communication that we observe at the network level. Experimental evidence relating pupil-linked arousal or neuromodulatory tone to the dynamic reorganization of network-level interaction is sparse. Recent data have shown that pharmacological blocking of noradrenaline reuptake in humans leads to a network-specific reduction in the correlation of hemodynamic signals between brain regions[47]. These findings are consistent with our results on the pupil-linked arousal-related reduction in thalamo-cortical spike correlations.

A compelling mechanistic description of how neuromodulation shapes network interactions has come from computational modeling and theoretic studies of complex networks. Most recently, Heeger[9] proposed a computational framework of cortical processing where network dynamics emerge from the interaction of feedforward and feedback inputs, as well as prior (expectation) driving factors. In this model, a number of state parameters modify the relative contribution of feedforward and feedback processing stages toward the predominating network dynamic. Our findings support Heeger's hypothesis that such state parameters represent the various neuromodulatory subsystems. Furthermore, Kirst et al.[48] showed that alteration of low-level features of complex networks not only leads to perturbations

in the collective dynamics of the network but can also result in the self-organized rerouting of information flow between network modules. Therefore, although neuromodulators act primarily on intrinsic cellular properties of neurons in cortex and thalamus, when translated to the network level, these local changes may lead to emergent shifts in thalamo-cortical functional interaction.

Although we studied the LP/Pulvinar–PPC thalamo-cortical network here, we speculate that our findings on arousal-dependent changes in interaction may generalize to other thalamo-cortical networks, and perhaps even modes of cortico-cortical functional interaction. One intriguing possibility is that subpopulations of neurons in the LC or basal forebrain differentially modulate the dynamics of specific sub-networks in the brain. Indeed, the heterogeneous nature of LC neuronal activity[49] coupled with non-overlapping projection patterns of ascending neuromodulatory fibers[50] may enable such network-specific modulation of functional interaction.

Extending these results further, we speculate that deficiencies in the neuromodulatory control of large-scale network interaction may represent a key mechanism underlying the pathology of various neuropsychiatric disorders[51]. A broader understanding of how neuromodulators shape functional interaction within affected brain networks will enable the targeted design of therapies aimed at restoring physiological patterns of network dynamics.

## Methods

**Animals**. Five adult spayed female ferrets (*Mustela putorius furo*) were used in this study. Animals had ad libitum access to food pellets and water, and were group housed in cages under standard ambient conditions (12 h day/light cycle). All animal procedures were performed in compliance with the National Institutes of Health guide for the care and use of laboratory animals (NIH Publications No. 8023, revised 1978) and the United States Department of Agriculture, and were approved by the Institutional Animal Care and Use Committee of the University of North Carolina at Chapel Hill.

**Headpost and electrode implantation**. Animals were initially anesthetized with an intramuscular injection of ketamine/xylazine (30 mg/kg of ketamine, 1–2 mg/kg of xylazine). After loss of the paw pinch reflex, animals were intubated to enable artificial ventilation and delivery of isoflurane anesthesia (0.5–2% isoflurane in 100% oxygen). Throughout surgical procedures, physiological parameters such as the electrocardiogram, end-tidal $CO_2$, partial oxygen concentration, and rectal temperature were constantly monitored to maintain the state of the animal. All surgical procedures were performed under sterile conditions. To enable the accurate planning of LP/Pulvinar electrode penetrations, animals were fixed into a stereotactic frame with stainless steel ear bars. The skull was then tilted such that it was oriented perpendicular to the surface of the surgical table. The skin and muscle were reflected to expose the surface of the skull. A custom-designed stainless steel headpost was firmly secured to the anterior extent of the exposed skull with bone screws. A craniotomy was performed on the left hemisphere to expose PPC[7] and the cortex overlying LP/Pulvinar. The dura was carefully removed, before a multielectrode array was lowered into LP/Pulvinar along stereotactic coordinates[52] (2 × 8 tungsten electrodes, 250 μm spacing, 9 mm length; Microprobes for Life Science). A second multielectrode array (4 × 8 tungsten electrodes, 200 μm spacing, Innovative Neurophysiology) was then inserted into the deep cortical layers of PPC. Both microelectrode arrays had reference electrodes that were directly adjacent to recording electrodes in the brain. In three animals, the PPC electrode was placed into the lateral gyrus, while in one animal the PPC electrode was placed in the suprasylvian gyrus. Multielectrode arrays were fixed in place with dental cement. Skin and muscle around the incision was then sutured together. After surgery, animals were administered preventative analgesics and antibiotics for one week. Animals were allowed to recover in their home cage for at least one week prior to recordings.

**Anatomical tracing experiments**. One adult female ferret was used for anatomical tracing. Preparation of the animal for aseptic surgery was performed according to procedures described above. Craniotomies were drilled above the PPC on both hemispheres and the dura was removed to expose the underlying cortical surface. A total of 0.8 μL of anterograde tracer (rAAV5-CaMKII-mCherry; UNC Vector Core) was injected between depths of 800 and 400 μm below the cortical surface in the left hemisphere. The location of the injection matched coordinates of cortical microelectrode array implantation in animals used for electrophysiology experiments (Supplementary Fig. 1). Retrograde tracer (0.8 μL cholera toxin subunit B conjugated to Alexa 488; Thermo Fisher Scientific) was injected at the corresponding location in the right hemisphere. Kwik-kast (World Precision Instruments) was

applied to the cortex to seal the craniotomy, before a layer of dental cement was applied to prevent regrowth of tissue over the injection sites. Three weeks following tracer injection surgery, the animal was humanely euthanized with an overdose of ketamine/xylazine, and then perfused with 0.1 M phosphate-buffered saline (PBS) initially, followed by 4% paraformaldehyde solution in 0.1 M PBS. After 2 days of fixation in 4% paraformaldehyde, the brain was transferred to 30% sucrose in 0.1 M PBS solution for cryoprotection. The brain was then sectioned into 50 μm slices using a cryostat (CM3050S; Leica Microsystems). Sections were imaged on a Nikon Eclipse 80i widefield microscope, with green, red, and brightfield images overlaid to construct composite images to illustrate fluorescent labeling in PPC and LP/Pulvinar.

**Recording procedure.** After recovery from surgery, animals were placed into a custom-designed behavioral tube and were head-fixed with a stainless steel head-post clamp. PPC and LP/Pulvinar multichannel recording electrodes were then connected to a data acquisition system (INTAN technologies). An infrared eye tracking camera was focused on the animal's right eye (ISCAN ETL-200, 240 Hz sample rate), with instantaneous measurements of pupil diameter, and pupil center $x/y$ position delivered as voltage signals to the data acquisition system. The infrared light source was positioned in the lower right of the animal's visual field and emitted non-visible light at a wavelength of 940 nm. Broadband extracellular potentials from both multielectrode arrays and eye tracking data were sampled at 30 kHz. Unless otherwise stated, recordings took place in a completely dark room under constant luminance (0.6 Fc). There was a dark adaptation period of at least 30 s prior to the start of each recording. To prevent animals from falling asleep while head-fixed, two experimenters engaged in verbal discourse in front of the animal. Recordings in the dark typically lasted between 8 and 13 min. In addition to recordings in the dark, data were collected also while animals passively viewed a library of 20 naturalistic images and 20 videos on a screen placed 28.5 cm in front of the animal. The eye tracking camera and infrared light source did not occlude the animal's view of any part of the screen. Images and videos were randomly interspersed with a stimulus duration of 10 s. A gray screen was presented during the interstimulus interval (interstimulus interval = 10 s; Fig. 5a). Visual stimuli were presented in a well-lit room (ambient luminance 32.6 Fc). Data were collected from multiple sessions across several days or weeks from each animal (number of sessions per animal: 6, 4, 7, 8).

**Verifying electrode positions with histology.** Animals were deeply anesthetized according to methods described above. Electrolytic lesions were then produced on both PPC and LP/Pulvinar probes by passing 5 μA of current (10 s pulse) between selected recording electrodes and the reference electrode on each probe. Animals were perfused and brains sectioned according to procedures described above. Brain slices were washed with 0.1 M PBS and stained for cytochrome c oxidase[53], and then imaged with a widefield microscope (Nikon Eclipse 80i; Nikon Instruments). Electrode positions in thalamus were confirmed by either the location of electrode tracks or electrolytic lesions. Electrodes that were deemed to fall outside of LP/Pulvinar were omitted from analysis (number of electrodes omitted per animal: 0, 2, 8, 6).

**Data analysis.** All offline data analyses were performed using custom software in Matlab (Mathworks).

Data preprocessing: To extract multi-unit spiking activity from broadband extracellular potentials, we band-pass filtered data between 300 and 5000 Hz and applied a threshold at −4 standard deviations. Spikes that were detected on more than three channels simultaneously were omitted from further analysis. LFPs were obtained by low-pass filtering broadband extracellular potentials at 300 Hz in both the forward and reverse direction to avoid phase shifts (fourth-order Butterworth filter). LFPs were then downsampled to a sample rate of 1 kHz. Eye position and pupil diameter signals were processed in the same way as the LFP.

Pupil diameter: To enable a thorough analysis of neuronal dynamics related to changes in pupil dilation, pupil diameter time series were discretized into eight bins of equal size, such that each bin represented 12.5% of samples from the entire recording (Fig. 2a). If the total number of samples per pupil bin did not exceed 30 s of cumulative data, then the recording was discarded ($n = 6$).

Saccade detection: Eye azimuth and elevation signals were low-pass filtered at 50 Hz to remove potential noise. Blinks were detected automatically, and data within ±1 s from blinks were removed from analysis. Elevation and azimuth time series were converted into eye velocity vectors. A threshold was then set at four standard deviations of the eye velocity vector to detect saccades. A subset of recording sessions in the dark were excluded from saccade based analyses due to high frequency noise in eye-position signals ($n = 12$).

Spectral decomposition: Time frequency estimates were computed by convolving LFP time series with Morlet wavelets that were Gaussian shaped in both the time and frequency domain[54,55].

$$w(t, f_0) = (\sigma_t \sqrt{\pi})^{-1/2} e^{-t^2/2\sigma_t^2} e^{-2\pi i f_0 t},$$

where $w(t, f_0)$ is the complex Morlet wavelet at carrier frequency $f_0$, and $\sigma_t$ is the

standard deviation of the wavelet in the time domain. $\sigma_t$ is defined as

$$\sigma_t = \frac{1}{2\pi\sigma_f},$$

where $\sigma_f$ is the standard deviation in the frequency domain, and is defined as a constant depending on the wavelet carrier frequency $\sigma_f = f_0/7$. Time frequency estimates $X(t, f_0)$ were then computed by convolving LFP time series $x(t)$ with complex Morlet wavelets $w(t, f_0)$:

$$X(t, f_0) = x(t) * w(t, f_0),$$

where $*$ denotes the convolution operation. We convolved LFP signals with a family of 80 Morlet wavelets that had carrier frequencies logarithmically spaced between 0.5 and 128 Hz. The power of LFP signals across all carrier frequencies was computed by taking the absolute value of squared time frequency estimates.

LFP phase synchronization: To quantify phase synchronization between simultaneously recorded LFP signals, we computed the phase locking value[56] (PLV). Briefly, the phase angle between real and complex components of time frequency estimates was calculated for thalamic $\theta^t$ and cortical $\theta^c$ signals. PLV at the carrier frequency $f_0$ for $N$ samples was then defined by the following formula:

$$PLV_{tc}(f_0) = \frac{1}{N} \left| \sum_{n=1}^{N} e^{i(\theta_n^t(f_0) - \theta_n^c(f_0))} \right|.$$

Assuming a uniform circular distribution of phases, the magnitude of $PLV_{tc}$ is biased towards 1 with few observations, and towards 0 with many observations. To control for this bias, pupil diameter-based PLV analyses were computed using a constant number of randomly drawn phase angles between conditions and animals (30,000 samples equating to 30 s of data). This was repeated for each thalamo-cortical channel pair 100 times, with the mean $PLV_{tc}$ across all permutations used for further analysis. For each recording session, the average thalamo-cortical $PLV_{tc}$ between all PPC and LP/Pulvinar channel pairs was calculated, and then used to compute the across-session average PLV.

One potential confound that arises when comparing PLV analyses between conditions with varying spectral power (e.g., alpha/theta power across pupil diameter bins) is that significant differences in PLV may emerge spuriously due to changes in the signal to noise ratio. To control for any potential signal to noise confounds for PLV estimates, we performed additional analyses where data were subsampled to match the power distributions across pupil diameter bins for each frequency band (Supplementary Fig. 4).

Spike–LFP phase synchronization: To determine the dependence of spike timing on LFP oscillation phase, we computed the PLV[56] between co-recorded spikes and LFPs both within and between brain regions. For within region analysis, spike PLV was computed between all channel combinations on each microelectrode array. Spike PLV analysis was not computed using the LFP on the same electrode to avoid spectral bleeding of spike waveforms contaminating phase synchronization estimates. For between region analyses, spike PLV was computed for all thalamo-cortical channel pairs. The spike PLV at a carrier frequency of $f_0$ for $K$ spikes was defined as

$$PLV_{spike}(f_0) = \frac{1}{K} \left| \sum_{k=1}^{K} e^{i\theta_k^{spike}(f_0)} \right|,$$

where $\theta^{spike}$ represents the instantaneous LFP phase at the occurrence of each spike. Spike PLVs were computed from 200 randomly drawn spikes for each channel pair combination. This process was repeated 40 times to obtain an estimate of the mean spike PLV. Channels with less than 200 spikes were detected for each pupil diameter bin were omitted from spike PLV analysis.

Granger causality: We computed Granger causality to measure the directed influence thalamic LFPs have on cortical LFPs, and vice versa. Granger causality is rooted in the autoregressive (AR) modeling of time series, where future values of a process $x(t)$ are modeled based on previous values of $x(t)$. In this framework, a separate process $y(t)$ can be said to have a causal influence on $x(t)$ if the past values of $y(t)$, when accounted for in a bivariate AR model, improve the prediction of future values of $x(t)$ beyond that obtained by the univariate AR model of $x(t)$ alone[57]. While classically employed in the time domain, the computation of Granger causality can be operationalized in the frequency domain to uncover the physiological carrier frequencies of directed interaction between brain regions[58]. We computed Granger causality between co-recorded LP/Pulvinar and PPC LFP signals in the frequency domain using the multivariate Granger causality (MVGC) toolbox for Matlab[59]. LFP signals from both regions were low-pass filtered at 100 Hz with a phase preserving filter, and then downsampled to 200 Hz. To reduce dimensionality, we computed representative signals from thalamus and cortex by calculating the median downsampled LFP signal across all channels in PPC and LP/Pulvinar microelectrode arrays. These signals were then windowed into segments of one second (200 samples) length. Model order was then selected based on the minimum Akaike information criterion value[59], with a maximum model order of 20 allowed. Vector AR-models were then checked to ensure that they reliably captured the spectral content of input data. Spectral Granger causality was

then computed according to routines from the MVGC toolbox. A random permutation test was used determine the significance of Granger causality peaks for individual recording sessions. Data segments from one brain region were randomly shuffled in a procedure that maintained spectral content, while disturbing the temporal codependence of co-recorded LFP signals. Granger causality measured on shuffled data represented the directed interaction that arises by chance based on the spectral signatures of the underlying neural processes. This procedure was repeated 100 times to generate a distribution of thalamo-cortical causal influence expected by chance. The distance of original Granger causality estimates from the shuffled mean were expressed in terms of standard deviation of the shuffled distribution, with values larger than 3 indicating significant Granger causal influence ($P < 0.01$). For Granger causality analysis based on pupil diameter, recordings were split into segments representing the lowest 25% or highest 25% of the pupil diameter distribution (Fig. 3b). Only data segments where pupil diameter was maintained in low or high states for more than one second were included. To control for signal to noise confounds, we subsampled data to match power distributions between low and high pupil diameter conditions (Supplementary Fig. 8). Power subsampling was performed based on the power of the predominant carrier frequency of thalamo-cortical causal influence in each direction (theta for LP/Pulvinar to PPC, alpha for PPC to LP/Pulvinar). To detect significant differences in the strength of Granger causality between low and high pupil diameter conditions, we performed $t$-tests on identified spectral peaks from data pooled across all recordings. In addition to pupil diameter linked granger causality, we computed time-resolved Granger causality around the occurrence of saccades with a sliding window of 1 s length and a step size of 0.25 s. To determine the significance of saccade related changes in Granger causality, we recomputed Granger causality at random time points throughout each recording session, where the number of data segments is equal to the number of detected saccades. This was repeated 1000 times for each recording session. Saccade-triggered Granger causality spectra were then normalized by the mean and standard deviation of randomly computed Granger causality spectra. Significant time and frequency points were those that deviated from the mean of randomly computed Granger causality estimates by 2 standard deviations ($P < 0.05$).

Phase slope index: Phase slope index (PSI) analysis is a non-autoregressive model based method that was used to quantify effective connectivity between co-recorded thalamic and cortical LFP signals. This form of analysis is grounded in the idea that if one brain region drives another brain region with a constant time lag, then one might expect the relative phase lag between signals in each region to increase as a function of carrier frequency[60]. PSI analysis was computed as described previously[60]. Briefly, LFP time series were windowed into segments of 1024 samples and the Fast Fourier Transform (FFT) on each data segment was computed. The complex coherency between thalamic and cortical signals at a carrier frequency of $f_0$ was then defined as

$$C_{tc}(f_0) = \frac{S_{tc}(f_0)}{\sqrt{S_{tt}(f_0)S_{cc}(f_0)}},$$

where $S_{tc}(f_0)$ represents the cross spectra, or the Fourier transform of thalamic signals $\hat{y}_t$ multiplied with the complex conjugate of the Fourier transform of cortical signals $\hat{y}_c^*$

$$S_{tc}(f_0) = \langle \hat{y}_t(f_0)\hat{y}_c^*(f_0) \rangle$$

$\langle \rangle$ indicates the computation of the expectation value. The phase slope is then computed across the frequency band of interest $f \rightarrow F$ as follows:

$$\Psi_{tc} = \mathrm{Imag}\left( \sum_f^F C_{tc}^*(f)C_{tc}(f + \delta f) \right),$$

where $*$ denotes the complex conjugate and $\delta f$ denotes the $(f+1)$th resolved frequency of the Fourier transform. The standard deviation of $\Psi_{tc}$ was estimated using the jackknife resampling method. Finally, $\Psi_{tc}$ was normalized by the standard deviation to estimate significance of effective thalamo-cortical connectivity, with values above 2 indicating that thalamus significantly drives cortex, and values below $-2$ indicating that cortex significantly drives thalamus ($P < 0.05$). We initially computed PSI using a sliding bandwidth of 4 Hz in the frequency domain to determine the carrier frequencies of thalamo-cortical effective connectivity. Since we observed both significant drivers and receivers across multiple frequency bands, the standard deviation of normalized PSI values was used to quantify the spread of reciprocal thalamo-cortical effective connectivity in the frequency domain. After identifying that the theta and alpha bands represent the carrier frequencies of thalamo-cortical effective connectivity, PSI values were recomputed using a frequency resolution of 2.5–6 and 11–18 Hz for theta and alpha bands, respectively. Finally, as a control measure PSI was additionally computed for the gamma band (30–60 Hz).

Spike–spike correlation: Spike cross-correlation analysis was used to determine the temporal dependence of co-recorded spiking activity in LP/Pulvinar and PPC. Spike time series in thalamus and cortex were binarized at a sample rate of 1 kHz. Cross-correlation functions were then computed on binarized time series for all possible combinations of LP/Pulvinar and PPC channel pairs. To determine the oscillatory structure of spike cross correlations, an FFT was computed on spike

cross-correlation functions from $-0.5$ to 0.5 s, with oscillatory power defined as the absolute value of the square of the FFT at each carrier frequency. To determine the dependence of synchronized oscillations in spiking on pupil diameter, spike cross-correlation functions were computed for spikes occurring during each pupil diameter bin. As above, the oscillatory power of pupil diameter dependent spike correlations was computed using an FFT for all pupil bins.

Statistics: All statistical tests were performed in Matlab (Mathworks). To test if neurophysiological and functional connectivity metrics significantly vary with fluctuations in pupil diameter, we computed a one-way ANOVA across the eight pupil diameter bins. For frequency resolved analyses, one-way ANOVAs were computed across the entire frequency spectrum, with $P$-values corrected for multiple comparisons using false discovery rate (Matlab function mafdr.m, with 'BHFDR' set to 'true'). To quantify the linear relationship between neural dynamics and changes in pupil diameter, we computed the Pearson correlation of neuronal and functional connectivity variables across the eight pupil diameter bins.

**Data availability**. Electrophysiological and pupillometry data, as well as MATLAB code that was used to perform outlined analyses, can be made available from the corresponding author upon request.

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

## Acknowledgements

We thank V. Crunelli, T. Pfeffer, A. Urai, K. Sellers, and Y. Li for comments on the manuscript; C. Yu for technical advice on obtaining recordings from LP/Pulvinar in ferrets; and G. Nolte for advice on data analysis.

## Author contributions

I.S. and F.F. conceived and designed experiments of state-dependent network dynamics; Z.C.Z., I.S., and F.F. conceived and designed experiments of visual processing and saccades; I.S. and Z.C.Z. performed implantation of microelectrode arrays, recorded data, and performed anatomical tracing experiments. S.R.S interpreted anatomical data. I.S. analyzed pupil and electrophysiological data. I.S., Z.C.Z., and F.F. wrote the paper. All authors contributed to the discussion of the results and revision of the manuscript.

## Additional information

**Competing interests:** The UNC has filed provisional patents on brain stimulation technology with F.F. as the lead inventor. No licensing has occurred. F.F. is the founder and majority shareholder of Pulvinar Neuro LLC. The work presented here has no relationship except the company is named after the senior author's favorite brain structure. The remaining authors declare no competing interests.

