## [Peer Review File · Nature Communications]

Reviewers' comments:

Reviewer #1 (Remarks to the Author):

The manuscript by Stitt et al. sets to tackle an important topic in systems neuroscience: how interactions between the thalamus and cortex change as a function of behavioral state. This topic is timely given the explosion of recent papers showing non-relay functions for the thalamus and re-interpretation of previous literature on the topic in light of such observations. The authors focus on the pulvinar, a higher-order thalamic nucleus that may be involved in regulating functional interactions across the visual stream as well as frontoparietal networks to which some of its divisions are reciprocally connected with. In fact, the authors focus on the pulvinar's subdivision that may be reciprocally-connected with the posterior parietal cortex (PPC). Experiments are done in ferrets, where the known role of PPC in primate spatial attention and decision making is not understood and therefore the overall relevance of the preparation to a specific cognitive process (like visuospatial attention or decision making) is unclear, but that's a minor issue. The authors collect spiking and lfp data from these regions (pulvinar and PPC), examine basic data properties (rates and spectral power) covariation with arousal, and then various correlations across these measures at different arousal levels (basically, behavioral epochs defined by binning the pupil diameter). Out of these analyses, they focus on one effect: directional interactions have distinct spectral features and are differentially modulated by arousal state. More specifically the thalamo-cortical signal is maximal at 4Hz and enhanced by arousal, while the cortico-thalamic signal is maximal at 16Hz and diminished by arousal. Showing videos to the animal enhances these effects and triggers eye movements. Frequency of eye movements is correlated with pupil diameter, and is therefore another putative measure of arousal in these animals and under the experimental conditions.

This is a narrow and descriptive study that neither provides mechanistic clarity nor computational meaning. I sincerely wished that it was different, as the authors seemed to have put in some genuine effort in collecting the data and making some sense of what they observed. However, in my view, problems with the experimental approach, data analysis style and interpretation make this paper inappropriate for publication at all as it is completely outside of the standards of the field. With moderate revisions and rewriting, the paper could be appropriate for a specialty journal. However, a completely different experimental (and intellectual) approach would be required for allowing the authors to translate their experimental preparation into data that would advance the field broadly, and would make sense to get out in a journal like Nature Communications. Below, I will make clarifications and provide feedback that I hope would help the authors do both: salvage this current paper and perform studies that advance the field in the future.

The main conceptual problem with this paper is that the abstract and introduction emphasize the role of thalamo-cortical interactions in cognitive computations. The experiments are done in head-fixed ferrets that are nodding off outside of a behavioral context and correlative measures across two broadly connected regions of a thalamo-cortical loop are being interpreted as cognitively-relevant. I found this to be quite frustrating, honestly, because I couldn't initially believe that that was all there was to the story. I kept looking for things in the manuscript that would make sense in light of the introduction and ended up wasting quite a bit of time doing so. I hope that this dissonance is reflective of the authors' naiveté, rather than scientific disingenuousness by overselling a limited set of observations as a broad advance. In fact, I am inclined to go with the former hypothesis as the authors' interpretation of their video presentation experiments cannot be anything other than naive; the animal is being shown a bunch of videos outside of a task context, an intervention that seems to be broadly arousing. Both the frequency of eye movements is increased as well as pupil diameter. These measures are correlated. Why exactly is this related to 'visual processing'? Do we even know that any of the recorded neurons are visually-responsive or care about any of the specific stimuli

shown? Now one thing that can be interesting (which sadly the authors do not do) is to understand whether the thalamo-cortical interactions observed are specifically related to eye movements and that the arousal changes they describe are really just bleed-through saccades. This is relatively simple, the authors could take behavioral epochs in which pupil diameter happens to be 87.5-100% and remove the peri-saccade components, recompute all the various measures and see if the 'arousal' effect goes away. Or the authors could try whatever approach to figure this out: because eye movements are correlated with arousal, it's important to understand whether the changes in directional interactions are related to the former or not.

Ideally, this would all be done in a behavioral context where the animal is being trained to move its eyes towards a behaviorally-relevant visual stimulus. I don't know how easy that is to do in ferrets, but the authors can do this in a closed-loop manner triggered on eye position (if the animal cannot fixate), and reward the animal for moving its eyes to the stimulus location. Of course, identifying the receptive fields of the recorded neurons in both thalamus and cortex would be important under such conditions and performing analyses related to whether the animal moves its eyes to or away from the receptive fields of a recorded neuron would also be critical. Identifying neurons using antidromic stimulation to make sure that an individual neuron is connected to the area of interest would also be important. These points are clearly well-beyond the current study, but these are the sort of approaches and analyses that will get at cognition and computations. Plus, these are not expectations above standard practice; these are approaches that have been performed in the eye movement field for the last three decades by Sommer, Wurtz, Goldberg and others, and have laid the foundation for our understanding of eye movement control (curiously, the authors do not cite any of these papers!).

Another major conceptual issue is that lack of a clear idea of what the pulvinar is doing to PPC and vice versa. Assuming that the authors have antidromically identified neurons from both structures, what do the authors think spiking in one area means for the other? Is the pulvinar relaying signals from the superior colliculus related to peri-saccade activity in the PPC? Is the pulvinar broadly enhancing excitability in the PPC? Is the pulvinar changing the connectivity patterns among PPC neurons (as some of the recent rodent papers showed)? Despite the limitations of their experimental preparation (the absence of antidromic identification and retinotopic mapping) the authors could potentially look at the relationship between spiking of individual neurons in pulvinar and local spiking cross-correlations in PPC (assuming local spiking is retinotopically similar in the ferret). This, they can do with the data they currently have. In the future, it would be nice to do some muscimol inactivation in the pulvinar and see whether any of the saccade related effects are truly causally related to pulvinar input (after all, granger causality isn't causal at all as the author state, it's just a directional correlation with a predictive temporal feature that can easily be related to a common causal input). Muscimol inactivation in the SC would also be important in that context. It's generally quite surprising why the authors don't do any manipulation that would allow for enhancing causal inference; you think pupil dilation is related to saccade behavior in some specific way? Inject a paralytic agent into the oculomotor muscles of one eye, both eyes. Put some atropine in the eye. See how these relationships change. Just make some attempt towards understanding the meaning of your findings rather than perform more correlations, and then more correlations of correlations.

Related to all of the above, the authors seem to have a high admiration for the work of Saalman, Kastner et al., but make no attempt in utilizing their approaches to behavioral control or examination of coordination of visual processing across multiple cortical areas. Just a little weird that this was the literature the authors chose to focus on citing rather than the eye movement literature given their findings (plus, why are most subsequent saccade analyses buried in the supplement?)

Other issues:

1. It is disingenuous to double dip with data presentation. Maybe the authors don't realize this, I don't know, but people don't show the exact same data in multiple locations (in some sense it's self-plagiarism). Take a look at Figure 1c and lower panel 1d, that's the same data as Supplemental Figure 1b. Just don't do that...

2. What's the meaning of showing Figure 2b when the underlying data is not normally-distributed (supplemental figure 2a)? Why can't you simply put replace Figure 2b with that?

3. In general the main paper is quite thin: have the authors simply not seen enough Nature Communications papers? Main Figure 1 is basically a non-figure (given that the data is basically a duplication of the supplemental figure 1), Figure 4 is a three panel figure, as is Figure 7. The authors would be wise to consider reorganizing their manuscript in a manner that bolsters their main presentation rather than have 13 supplemental figures. Also, would be good to show some more raw data; hard to know what any of the spiking data mean without getting a sense of which spikes are related to what events and why the authors are getting the high-level measures they are getting. Overall data transparency is just low in this work.

Summary:

I definitely encourage the authors to perform more detailed analysis of their spiking data to get a sense of the impact of pulvinar on PPC. Might be hard without antidromic identification to be sure, but showing the data on a cell by cell basis (e.g. distribution of the following measure: PPC spiking cross-corr/noise corr triggered on pulvinar spiking (spike times, thresholded rate, bursts)). Such measures would probably provide some computational insight into what this thalamo-cortical pathway actually does. Clearly, pharmacological inactivation is necessary, muscimol in pulvinar, and muscimol in superior colliculus (two separate experiments). This, along with what the authors already have may allow for some advance in the field but unlikely to be of broad interest to the general neuroscientist. Adding a behavioral context, and controlling eye movements could. Please take my advice constructively, if I didn't want to help, I would have given this the time of day.

Reviewer #2 (Remarks to the Author):

This is a much enhanced manuscript with only a few aspects left deserving adjustment.

Overall it is acknowledged that the authors added substantial new data and results including important ones like (1) histological evidence for anatomical connections between pulvinar and PPC, (2) clarification of the relationship of spiking activity and pupil changes and of the overall firing rates in pulvinar/parietal cortex, (3) revised statistical approaches for granger causality and PLV analysis, and (4) various other clarifications such as the justification of the definition of the frequency labels and methods aspects that were unclear in the first submission.

The abstract is rather vague in describing the main findings by stating only "...changes in the direction and carrier frequency of oscillatory interaction..." between PPC and pulvinar. This should be made more specific.

The saccade- aligned granger causality analysis is very interesting, but the presentation of the results should include more specific information about the timing of the observed thea granger increase and alpha reductions relative to saccade onset. Fort both effects the figures shows that significant effects already starts prior to the saccade and then last much longer than the average inter-saccade time. Is

the anticipatory effect an artifact of the long 1 sec time window? Are the post-saccadic granger reductions (alpha) and increases (theta) influenced by saccades occurring shortly after the alignment saccade.

I think these questions should either be addressed directly, or the authors should add several sentences clarifying that the time resolved analysis is influenced by saccades occurring before and after the alignment saccade and how these other saccades might affect the overall general result that the author want to present.

Reviewer #3 (Remarks to the Author):

Stitt, Zhou Radtke-Schuller & Fröhlich: "Arousal dependent modulation of thalami-cortical functional interaction"

[Redacted] I am glad to see that the authors have addressed all the main concerns I had raised about the previous version, with additional experiments (the visual stimulation condition), several analyses (e.g., power-matched versions of interaction measures), and substantial re-writing (eliminating speculations about bottom-up vs. top-down signal flow, as well as toning down speculations about function). Below, I am re-iterating my assessment of the paper and its potential impact, taking into account all the changes the authors have made since then.

This study is on a very timely topic. Overall, it is carefully executed, and technically sophisticated, and the manuscript clearly written. The central claim of the paper is that pupil-linked arousal (and by extension neuromodulatory tone) controls the structure of functional interactions between thalamus and cortex of the ferret. Specifically, the claim is that cortex drives the pulvinar nucleus of the thalamus when pupil diameter is small; by contrast, pulvinar drives cortex when pupil diameter is large. These differences in information flow between brain areas are evident in differential patterns of (i) local LFP power spectra, (ii) thalamocortical phase synchrony, and (iii) directed interactions between both structures, assessed through Granger causality and phase slope measures. Corresponding changes in network interaction measures take place under visual stimulation (especially movies) and seem to be coupled to oculomotor exploration of the input.

The paper will likely meet large interest by a broad range of neuroscientists working at different levels of brain organisation: neurophysiologists and theoretical neuroscientists studying the impact of neuromodulation and arousal state on cortical network dynamics, neuroimaging researchers / systems neuroscientists characterising brain-wide patterns of "resting-state" activity (i.e., co-variations of ongoing activity in different brain regions), and cognitive neuroscientists / psychologists interested in the physiological basis of cognitive pupillometry.

The study is the first to comprehensively characterize pupil-linked changes in the *interactions* between brain regions. The paper's main limitation is that all measurements are made without a controlled behavioural task, and the case for meaningful behavioural consequences remains weak. Even so, the reported changes in the interaction measures are complex and striking, novel, and remarkably compelling. Thus, I expect that the paper will have significant impact on a rapidly developing field of neuroscience, and I believe it warrants publication in Nature Communications, pending revisions. I would like the authors to address the following remaining issues prior to publication.

1. The logic of the current conclusion seems to be the following: when comparing large vs. small

diameter periods during rest, the spectral profiles of the differences in LFP power and interaction measures are similar to the stimulus-induced changes of the same measures (Fig. 5). While I find this largely compelling by looking at the data, the conclusion would be stronger if a quantitative measure of similarity was used. For example, by correlating the spectra of the modulations induced by large pupil diameter and visual stimulation within animals and the strength of these modulations across animals. The same applies to the saccade-related modulations shown in Fig. 6.

2. Given the general interest in NE-modulation and the recent findings by Reimer and colleagues: Why not perform all the analyses based on binning of the pupil derivative time series, in addition to the raw pupil time series? Especially qualitative differences in the neurophysiological correlates of both might be highly informative.

3. Figure 6 e-g and the accompanying conclusions about the modulations of interaction measures around saccades seem to lack statistical assessments. Please clarify.

4. Two further studies in addition to the one by Joshi et al showed a correlation between brainstem/LC-activity and pupil diameter, both using advanced fMRI approaches: Murphy et al, Human Brain Mapping, 2014; de Gee et al, eLife, 2017. Both studies should be cited in addition to the monkey and rodent work.

5. The latter study also shows correlations in the basal forebrain (as well as dopaminergic structures and the superior colliculus) with transient pupil dilation amplitude. So the interpretation of the phasic dilations as pure readout of NE-release should be toned down, despite the (very nice) findings by Reimer et al.

Author's response to reviewers

We thank all three reviewers for the thorough feedback we received on this manuscript. We recognize how much time and effort all three reviewers have spent to help us improve our manuscript. Please find below our detailed answers. For your convenience, we have labeled each reviewer comment and our response. Thank you for your time and input.

Reviewer # 1:

The manuscript by Stitt et al. sets to tackle an important topic in systems neuroscience: how interactions between the thalamus and cortex change as a function of behavioral state. This topic is timely given the explosion of recent papers showing non-relay functions for the thalamus and re-interpretation of previous literature on the topic in light of such observations. The authors focus on the pulvinar, a higher-order thalamic nucleus that may be involved in regulating functional interactions across the visual stream as well as frontoparietal networks to which some of its divisions are reciprocally connected with. In fact, the authors focus on the pulvinar's subdivision that may be reciprocally-connected with the posterior parietal cortex (PPC). Experiments are done in ferrets, where the known role of PPC in primate spatial attention and decision making is not understood and therefore the overall relevance of the preparation to a specific cognitive process (like visuospatial attention or decision making) is unclear, but that's a minor issue. The authors collect spiking and lfp data from these regions (pulvinar and PPC), examine basic data properties (rates and spectral power) covariation with arousal, and then various correlations across these measures at different arousal levels (basically, behavioral epochs defined by binning the pupil diameter). Out of these analyses, they focus on one effect: directional interactions have distinct spectral features and are differentially modulated by arousal state. More specifically the thalamo-cortical signal is maximal at 4Hz and enhanced by arousal, while the cortico-thalamic signal is maximal at 16Hz and diminished by arousal. Showing videos to the animal enhances these effects and triggers eye movements. Frequency of eye movements is correlated with pupil diameter, and is therefore another putative measure of arousal in these animals and under the experimental conditions.

Author's Response:

We thank Reviewer one for the interest in the topic of our study and the careful summary. With regards to choice of species, we would like to propose that the fact that we performed our study in ferrets instead of non-human primates adds novelty to our study.

Reviewer # 1:

This is a narrow and descriptive study that neither provides mechanistic clarity nor computational meaning. I sincerely wished that it was different, as the authors seemed to have put in some genuine effort in collecting the data and making some sense of what they observed. However, in my view, problems with the experimental approach, data analysis style and

interpretation make this paper inappropriate for publication at all as it is completely outside of the standards of the field. With moderate revisions and rewriting, the paper could be appropriate for a specialty journal. However, a completely different experimental (and intellectual) approach would be required for allowing the authors to translate their experimental preparation into data that would advance the field broadly, and would make sense to get out in a journal like Nature Communications. Below, I will make clarifications and provide feedback that I hope would help the authors do both: salvage this current paper and perform studies that advance the field in the future.

Author's Reply:

We appreciate the Reviewer's efforts. We worked hard to address the substantive points raised. Please see below for details. We respectfully disagree with the assessment of our study as "narrow and descriptive." To our knowledge (please also see the assessment by Reviewer 3), our work provides a substantial step forward since it is the first to delineate how rhythmic structure shapes communication in the thalamo-cortical system as a function of arousal. Importantly, our results are embedded in the context of active vision by saccadic sampling during presentation of naturalistic video clips.

Reviewer # 1:

Ideally, this would all be done in a behavioral context where the animal is being trained to move its eyes towards a behaviorally-relevant visual stimulus. I don't know how easy that is to do in ferrets, but the authors can do this in a closed-loop manner triggered on eye position (if the animal cannot fixate), and reward the animal for moving its eyes to the stimulus location. Of course, identifying the receptive fields of the recorded neurons in both thalamus and cortex would be important under such conditions and performing analyses related to whether the animal moves its eyes to or away from the receptive fields of a recorded neuron would also be critical. Identifying neurons using antidromic stimulation to make sure that an individual neuron is connected to the area of interest would also be important. These points are clearly well-beyond the current study, but these are the sort of approaches and analyses that will get at cognition and computations. Plus, these are not expectations above standard practice; these are approaches that have been performed in the eye movement field for the last three decades by Sommer, Wurtz, Goldberg and others, and have laid the foundation for our understanding of eye movement control (curiously, the authors do not cite any of these papers!).

Author's Reply:

The reviewer proposes some interesting directions for future research. We also agree with the reviewer that these studies are beyond the current submission. In fact, the kind suggestions by the Reviewer equate to a research program for many years if not decades. We appreciate all the great ideas and will make sure to include them into our research program. We would like to emphasize that our study focuses on mesoscale dynamics of the local field potential, which represents a distinct approach from the classical literature in non-human primates that focuses on microscopic dynamics of single unit firing.

Reviewer # 1:

Another major conceptual issue is that lack of a clear idea of what the pulvinar is doing to PPC and vice versa. Assuming that the authors have antidromically identified neurons from both structures, what do the authors think spiking in one area means for the other? Is the pulvinar relaying signals from the superior colliculus related to peri-saccade activity in the PPC? Is the pulvinar broadly enhancing excitability in the PPC? Is the pulvinar changing the connectivity patterns among PPC neurons (as some of the recent rodent papers showed)? Despite the limitations of their experimental preparation (the absence of antidromic identification and retinotopic mapping) the authors could potentially look at the relationship between spiking of individual neurons in pulvinar and local spiking cross-correlations in PPC (assuming local spiking is retinotopically similar in the ferret). This, they can do with the data they currently have. In the future, it would be nice to do some muscimol inactivation in the pulvinar and see whether any of the saccade related effects are truly causally related to pulvinar input (after all, granger causality isn't causal at all as the author state, it's just a directional correlation with a predictive temporal feature that can easily be related to a common causal input). Muscimol inactivation in the SC would also be important in that context. It's generally quite surprising why the authors don't do any manipulation that would allow for enhancing causal inference; you think pupil dilation is related to saccade behavior in some specific way? Inject a paralytic agent into the oculomotor muscles of one eye, both eyes. Put some atropine in the eye. See how these relationships change. Just make some attempt towards understanding the meaning of your findings rather than perform more correlations, and then more correlations of correlations.

Author's Reply:

Again, we greatly appreciate the suggested directions for future research. For the current submission, we have focused on clarifying and addressing the questions about the data presented. For the Granger causality, we agree that one can have long and enjoyable philosophical debates about the definition of causality. From a scientific viewpoint, we have performed several additional new directions of analysis to hone in on the link between arousal as indexed by pupil diameter (velocity) and network interactions, with focus on the rhythmic structure. We have refrained from additional correlative assessments to the extent possible to address the general concern of the reviewer that our results are too distant from the raw data. We also note that there are no convincing studies showing a retinotopic organization of PPC in ferrets, which would, as stated by the Reviewer, be a necessary condition for the additional analysis proposed.

Reviewer # 1:

Related to all of the above, the authors seem to have a high admiration for the work of Saalman, Kastner et al., but make no attempt in utilizing their approaches to behavioral control or examination of coordination of visual processing across multiple cortical areas. Just a little weird that this was the literature the authors chose to focus on citing rather than the eye movement literature given their findings (plus, why are most subsequent saccade analyses buried in the supplement?)

Author's Reply:

The paper by Saalman et al has been an inspiration for us to in terms of our focus on investigating the role of rhythmic organization of activity in the pulvinar of ferrets. We did not mean to offend!

Reviewer # 1:

The main conceptual problem with this paper is that the abstract and introduction emphasize the role of thalamo-cortical interactions in cognitive computations.

Author's response:

We understand this concern and have addressed this by adjusting the language accordingly. To avoid confusion and to narrow the context of this papers interpretation, we have toned down the references to cognitive computation in the abstract and introduction of the revised manuscript.

Reviewer # 1:

The experiments are done in head-fixed ferrets that are nodding off outside of a behavioral context and correlative measures across two broadly connected regions of a thalamo-cortical loop are being interpreted as cognitively-relevant. I found this to be quite frustrating, honestly, because I couldn't initially believe that that was all there was to the story. I kept looking for things in the manuscript that would make sense in light of the introduction and ended up wasting quite a bit of time doing so. I hope that this dissonance is reflective of the authors' naïveté, rather than scientific disingenuousness by overselling a limited set of observations as a broad advance.

Author's response:

We assure the Reviewer that we have no intention to oversell our results and that we respectfully disagree with the conclusion that we are naïve. We have presented this work at multiple international meetings in the fields of psychiatry, neurology, brain stimulation, sleep, and systems neuroscience and have consistently received very positive feedback. We are comfortable publishing results that are less complete or less innovative in lower ranked journals. However, we feel that based on the feedback we have received form the community and the other two reviewers that the results warrant publication in a journal with broader readership.

Reviewer # 1:

In fact, I am inclined to go with the former hypothesis as the authors' interpretation of their video presentation experiments cannot be anything other than naïve; the animal is being shown a bunch of videos outside of a task context, an intervention that seems to be broadly arousing. Both the frequency of eye movements is increased as well as pupil diameter. These measures are correlated. Why exactly is this related to 'visual processing'? Do we even know that any of the recorded neurons are visually-responsive or care about any of the specific stimuli shown? Now one thing that can be interesting (which sadly the authors do not do) is to understand whether the thalamo-cortical interactions observed are specifically related to eye movements and that the arousal changes they describe are really just bleed-through saccades. This is relatively simple,

the authors could take behavioral epochs in which pupil diameter happens to be 87.5-100% and remove the peri-saccade components, recompute all the various measures and see if the 'arousal' effect goes away. Or the authors could try whatever approach to figure this out: because eye movements are correlated with arousal, it's important to understand whether the changes in directional interactions are related to the former or not.

Author's response:

As Reviewer #1 suggested, we have reanalyzed our data after eliminating LFP data from peri-saccadic periods (± 1 s from saccades). These results can be found in Supplementary Figure 12. Briefly, the main structure of the functional connectivity and organization of spiking by the rhythmic organization of the LFP remain present.

Reviewer # 1:

1. It is disingenuous to double dip with data presentation. Maybe the authors don't realize this, I don't know, but people don't show the exact same data in multiple locations (in some sense it's self-plagiarism). Take a look at Figure 1c and lower panel 1d, that's the same data as Supplemental Figure 1b. Just don't do that...

Author's response:

The purpose of Supplementary Figure 1 is to illustrate that electrophysiological recordings were obtained from comparable regions in PPC and LP/Pulvinar that we have shown to be reciprocally connected. We have included the tracing images in the Supplementary Figure to allow for a side-by-side comparison, such that the reader does not have to flip pages back and forth to make the comparison themselves. We respectfully object to the suggestion of self-plagiarism.

Reviewer # 1:

2. What's the meaning of showing Figure 2b when the underlying data is not normally-distributed (supplemental figure 2a)? Why can't you simply put replace Figure 2b with that?

Author's response:

The two figures that Reviewer #1 is referring to actually show completely different information. Supplementary Figure 2a shows the overall distribution of firing rates in PPC and LP/Pulvinar (without any relationship to state). Whereas, Figure 2b is shows the mean firing rate in each pupil bin, which is normalized by the overall firing rate across the recording for each unit (where a normalized rate of 1 would equal the mean FR across the session). With respect to the underlying distribution of the firing rate modulations in Figure 2b, they are not perfectly normally distributed (as can be seen in the figure below), but they could be generally approximated by a normal distribution.

Reviewer # 1:

3. In general the main paper is quite thin: have the authors simply not seen enough *Nature Communications* papers? Main Figure 1 is basically a non-figure (given that the data is basically a duplication of the supplemental figure 1), Figure 4 is a three panel figure, as is Figure 7. The authors would be wise to consider reorganizing their manuscript in a manner that bolsters their main presentation rather than have 13 supplemental figures. Also, would be good to show some more raw data; hard to know what any of the spiking data mean without getting a sense of which spikes are related to what events and why the authors are getting the high-level measures they are getting. Overall data transparency is just low in this work.

Author's response:

We respectfully disagree with this comment. We feel that the number of panels per figure is not an appropriate measure to judge the quality of a scientific manuscript.

Reviewer #2:

This is a much enhanced manuscript with only a few aspects left deserving adjustment.

Overall it is acknowledged that the authors added substantial new data and results including important ones like (1) histological evidence for anatomical connections between pulvinar and PPC, (2) clarification of the relationship of spiking activity and pupil changes and of the overall firing rates in pulvinar/parietal cortex, (3) revised statistical approaches for granger causality and PLV analysis, and (4) various other clarifications such as the justification of the definition of the frequency labels and methods aspects that were unclear in the first submission.

Author's response:

We thank Reviewer #2 for appreciating all the new results we added on the previous round of revisions.

Reviewer #2:

The abstract is rather vague in describing the main findings by stating only "...changes in the direction and carrier frequency of oscillatory interaction..." between PPC and pulvinar. This should be made more specific.

Author's response:

We appreciate the feedback, and we have modified the abstract to be more specific, and more directly refer to some of the underlying phenomena we uncover in this work.

Reviewer #2:

The saccade- aligned granger causality analysis is very interesting, but the presentation of the results should include more specific information about the timing of the observed theta granger increase and alpha reductions relative to saccade onset. For both effects the figures shows that significant effects already starts prior to the saccade and then last much longer than the average inter-saccade time. Is the anticipatory effect an artifact of the long 1 sec time window? Are the post-saccadic granger reductions (alpha) and increases (theta) influenced by saccades occurring shortly after the alignment saccade.

I think these questions should either be addressed directly, or the authors should add several sentences clarifying that the time resolved analysis is influenced by saccades occurring before and after the alignment saccade and how these other saccades might affect the overall general result that the author want to present.

Author's response:

Reviewer #2 makes an important point (one that Reviewer #1 also pointed out). To address these issues we have expanded Supplementary Figure 11 to include spectrograms illustrating the time course of significant LFP power and PLV modulations with respect to saccades. Unlike Granger Causality analyses, which were computed in a sliding window of 1 second width, these analyses were based on wavelets, which have a more optimal time resolution and therefore provide greater insight into timing of spectral modulations around saccades.

In addition, to disentangle the contribution of saccades and pupil diameter fluctuations, we performed functional connectivity analyses after removing peri-saccadic epochs. This new analysis can be seen in Supplementary Figure 12 in the revised manuscript.

Reviewer #3:

[Redacted] I am glad to see that the authors have addressed all the main concerns I had raised about the previous version, with additional experiments (the visual stimulation condition), several analyses (e.g., power-matched versions of interaction measures), and substantial re-writing (eliminating speculations about bottom-up vs. top-down signal flow, as well as toning down speculations about function). Below, I am re-iterating my assessment of the paper and its potential impact, taking into account all the changes the authors have made since then.

This study is on a very timely topic. Overall, it is carefully executed, and technically sophisticated, and the manuscript clearly written. The central claim of the paper is that pupil-linked arousal (and by extension neuromodulatory tone) controls the structure of functional interactions between thalamus and cortex of the ferret. Specifically, the claim is that cortex drives the pulvinar nucleus of the thalamus when pupil diameter is small; by contrast, pulvinar drives cortex when pupil diameter is large. These differences in information flow between brain areas are evident in differential patterns of (i) local LFP power spectra, (ii) thalamocortical phase synchrony, and (iii) directed interactions between both structures, assessed through Granger causality and phase slope measures. Corresponding changes in network interaction measures take place under visual stimulation (especially movies) and seem to be coupled to oculomotor exploration of the input.

The paper will likely meet large interest by a broad range of neuroscientists working at different levels of brain organisation: neurophysiologists and theoretical neuroscientists studying the impact of neuromodulation and arousal state on cortical network dynamics, neuroimaging researchers / systems neuroscientists characterising brain-wide patterns of “resting-state” activity (i.e., co-variations of ongoing activity in different brain regions), and cognitive neuroscientists / psychologists interested in the physiological basis of cognitive pupillometry.

*The study is the first to comprehensively characterize pupil-linked changes in the *interactions* between brain regions. The paper’s main limitation is that all measurements are made without a controlled behavioural task, and the case for meaningful behavioural consequences remains weak. Even so, the reported changes in the interaction measures are complex and striking, novel, and remarkably compelling. Thus, I expect that the paper will have significant impact on a rapidly developing field of neuroscience, and I believe it warrants publication in Nature Communications, pending revisions. I would like the authors to address the following remaining issues prior to publication.*

Author's response

We thank Reviewer 3 for his/her enthusiasm for our submission. We greatly appreciate all the helpful comments on the previous and the current round of edits, which have greatly improved our manuscript.

1. The logic of the current conclusion seems to be the following: when comparing large vs. small diameter periods during rest, the spectral profiles of the differences in LFP power and interaction measures are similar to the stimulus-induced changes of the same measures (Fig. 5). While I find this largely compelling by looking at the data, the conclusion would be stronger if a quantitative measure of similarity was used. For example, by correlating the spectra of the modulations induced by large pupil diameter and visual stimulation within animals and the strength of these modulations across animals. The same applies to the saccade-related modulations shown in Fig. 6

Author's response:

We appreciate Reviewer #3's attention to detail with this comparison, and indeed have followed through with the suggestion to correlate LFP power modulations as a result of pupil diameter fluctuations, visual stimulation, and saccades. However, we did not correlate the strength of modulations across animals, since we only have 4 animals and thought this was too small a number to estimate correlation.

Here are the relevant excerpts from the revised manuscript:

“LFP power modulations during visual processing were significantly correlated with LFP power spectra during large pupil diameter states in the absence of any stimulus (PPC correlation per animal: 0.51*, 0.94*, 0.92*, 0.82*; LP/Pulvinar correlation per animal: 0.13, 0.66*, 0.06, 0.54*; * P < 0.001).”

“In line with this, PPC and LP/Pulvinar LFP power during saccades was significantly correlated with power spectra during large pupil diameter states (PPC correlation per animal: 0.95*, 0.99*, 0.91*, 0.94*; LP/Pulvinar correlation per animal: 0.71*, 0.91*, 0.81*, 0.88*; * P < 0.01), with power modulations occurring over a time course of several seconds around saccades (Figure 6e, P < 0.05, test against random saccade times, Supplementary Figure 11a-b).”

Reviewer #3:

2. Given the general interest in NE-modulation and the recent findings by Reimer and colleagues: Why not perform all the analyses based on binning of the pupil derivative time series, in addition to the raw pupil time series? Especially qualitative differences in the neurophysiological correlates of both might be highly informative.

Author's response:

We thank Reviewer #3 for this fantastic suggestion. Indeed, we reanalyzed all of our data based on the derivative of pupil diameter time series. This led to new and interesting findings that have

now been incorporated into the revised manuscript, with new results strongly complementing our previous findings. The bulk of these new findings can be found in Figure 8 of the revised manuscript.

Reviewer #3:

3. Figure 6 e-g and the accompanying conclusions about the modulations of interaction measures around saccades seem to lack statistical assessments. Please clarify.

Author's response:

We have included statistical tests for these figures in the revised manuscript.

Reviewer # 3:

4. Two further studies in addition to the one by Joshi et al showed a correlation between brainstem/LC-activity and pupil diameter, both using advanced fMRI approaches: Murphy et al, Human Brain Mapping, 2014; de Gee et al, eLife, 2017. Both studies should be cited in addition to the monkey and rodent work.

Author's response:

We have incorporated a discussion of these human studies in the relevant part of the discussion section.

Reviewer #3:

5. The latter study also shows correlations in the basal forebrain (as well as dopaminergic structures and the superior colliculus) with transient pupil dilation amplitude. So the interpretation of the phasic dilations as pure readout of NE-release should be toned down, despite the (very nice) findings by Reimer et al.

Author's response:

This is indeed a very interesting (and complex) issue. Accordingly, we have toned down our references to specific neuromodulatory subsystems in the revised manuscript.

REVIEWERS' COMMENTS:

Reviewer #2 (Remarks to the Author):

The updated manuscript constructively addressed all previous suggestions. I believe this could become an influential paper.

Reviewer #3 (Remarks to the Author):

The authors have done a great job in reviewing this manuscript, and addressed all my remaining concerns.

I realize that reviewer 1 has a quite different take on the relevance of this paper. Nonetheless, I maintain think the result are timely and important (for reasons provided in my initial review), and I am positive that the paper will attract quite some attention in the field.

I support publication in Nature Communication.